# TripletCLIP: Improving Compositional Reasoning of CLIP via Synthetic Vision-Language Negatives

**Maitreya Patel**$^{\diamond *}$  **Abhiram Kusumba**$^{\diamond \dagger}$  **Sheng Cheng**$^{\diamond \dagger}$  **Changhoon Kim**$^{\diamond}$
**Tejas Gokhale**$^{\heartsuit}$  **Chitta Baral**$^{\diamond}$  **Yezhou Yang**$^{\diamond}$
$^{\diamond}$Arizona State University   $^{\heartsuit}$University of Maryland, Baltimore County
tripletclip.github.io

## Abstract

Contrastive Language-Image Pretraining (CLIP) models maximize the mutual information between textual and visual modalities to learn representations. However, the lack of compositional diversity in contemporary image-text datasets limits the compositional reasoning ability of CLIP. We show that generating "hard" negative captions via in-context learning and synthesizing corresponding negative images with text-to-image generators offers a solution. We introduce a novel contrastive pre-training strategy that leverages these hard negative captions and images in an alternating fashion to train CLIP. We demonstrate that our method, named `TripletCLIP`, when applied to existing datasets such as CC3M and CC12M, enhances the compositional capabilities of CLIP, resulting in an absolute improvement of over 9% on the SugarCrepe benchmark on an equal computational budget, as well as improvements in zero-shot image classification and image retrieval. Our code, models, and data are available at: tripletclip.github.io.

## 1 Introduction

Large-scale vision-language models, such as CLIP [41], have significantly advanced multi-modal learning by employing contrastive learning to acquire shared semantic representations from paired datasets. This approach has resulted in improved performance in vision-language tasks as well as zero-shot image classification [51] and segmentation [23, 62]. Beyond vision-language tasks, the individual components of these models, such as the vision encoder and the language encoder, are integral to several multimodal architectures and generative models such as multimodal large language models (MLLMs) [30, 26] and text-to-image (T2I) diffusion models [45, 37, 38]. Yet, compositional reasoning remains challenging and multimodal models continue to exhibit naïve "bag of words" behavior, frequently failing to distinguish between expressions like "bulb in the grass" and "grass in the bulb" [58, 53]. Addressing this challenge remains critical for enhancing vision-language models and their downstream applications.

Contrastive learning of representations benefits from "hard negative samples" (i.e., points that are difficult to distinguish from an anchor point) [43]. However, at each optimization step for training CLIP, image-text pairs are *randomly* sampled from the training dataset – this random sampling seldom exposes the model to highly similar negative pairs. We hypothesize that the limited compositional understanding of CLIP may stem from such issues in the optimization objective and sampling from training datasets. A straightforward solution could involve iteratively identifying hard negative pairs for each training iteration. However, due to the noisy captions and the scarcity of such pairs in existing datasets, prior work generates hard negative captions as a form of augmentation using rule-based strategies [58, 61]. For instance, given an image-text pair labeled "a brown horse", an additional

---

$^{*}$Corresponding author: maitreya.patel@asu.edu. $\dagger$ indicates the equal contribution.

38th Conference on Neural Information Processing Systems (NeurIPS 2024).

negative caption "a blue horse" might be introduced. However in prior work, image data is not subjected to similar hard negative semantic augmentation during training; this is mainly because of the difficulty of making semantic perturbations at the pixel levels compared to sentence perturbation. While the text-only augmentation strategies have improved the models' compositional understanding to a certain extent, it raises an intriguing question: ***could incorporating hard negative augmentation for both text and image modality further enhance the compositional reasoning capabilities of vision-language models?***

Motivated by this, in this paper, we introduce a novel, simple, and yet highly effective strategy for integrating hard negative images as well as hard negative text to enhance the compositional understanding of vision-language models. Recent developments in text-to-image diffusion models have opened up possibilities for performing semantic perturbations within images [21]. Existing works have evaluated the impact of creating synthetic data for text-to-image generative models [3, 6]. However, it remains less explored how these generative models can benefit the CLIP-like models. To tackle this challenge, our approach leverages the in-context learning capabilities of LLMs to produce realistic, linguistically accurate negative captions [55]. We then employ a pre-trained text-to-image diffusion model to create images corresponding to these captions, thereby enriching any given image-text dataset with valuable hard negatives that foster improved reasoning. This resulting `TripletData` comprises 13M image-text pairs to complement the CC3M and CC12M datasets [5].

We developed `TripletCLIP`, which incorporates hard negative image-text pairs effectively by using them to optimize a novel triplet contrastive loss function. Extensive experiments on the CC3M and CC12M datasets and various downstream tasks with an equal compute budget demonstrate that `TripletCLIP` significantly enhances compositional reasoning. Notably, `TripletCLIP` results in more than 9% and 6% absolute improvement on the SugarCrepe benchmark compared to LaCLIP and NegCLIP, respectively. `TripletCLIP` also improves zero-shot classification and image-text retrieval performance with similar training-time concept diversity. An investigation into the effects of increasing training-time concept diversity revealed that baseline models consistently under-performed in compositional tasks despite an increase in integrated knowledge, while `TripletCLIP` demonstrated significant improvements. In summary, our **key contributions** are as follows:

- We introduce a novel CLIP pre-training strategy that employs hard negative images in conjunction with triplet contrastive learning to enhance compositionality.
- `TripletCLIP` consistently improves across downstream tasks, demonstrating the effectiveness of synthesizing hard negative image-text pairs.
- Our extensive ablations on the choice of the loss function, modality-specific pre-training, the increase in concept diversity, and filtering high-quality `TripletData` provide deeper insights into the utility of hard negative image-text pairs for CLIP pre-training.
- Ultimately, we present a promising avenue where synthetic contrastive datasets significantly improve reasoning capabilities, leading to the creation and release of the `TripletData` — a 13M contrastive image-text dataset.

## 2 Related Work

**Vision-Language Models.** Recent advancements, including ALIGN [22] and CLIP [41], have gained significant interest due to their capability to learn transferable semantic representations across multiple modalities through contrastive learning. These models facilitate downstream tasks such as zero-shot classification [51], image-text retrieval [60, 2], visual grounding/reasoning [31], text-to-image generation [52, 37, 38, 24], semantic segmentation [62, 23], and various evaluations [19, 47]. Subsequent research has sought to enhance various aspects of these models, including data efficiency [15], hierarchical representation learning [8], and the quantization of latent spaces for more stable pre-training [7]. LiT [59] employs a pre-trained frozen CLIP vision encoder to fine-tune a BERT-like text encoder [9], achieving notable improvements in zero-shot transfer performance. Similarly, BLIP-2 [27] combines contrastive pre-training with the next-token prediction for image captioning during training. However, these approaches generally presume the availability of high-quality data. In contrast, `TripletCLIP` focuses on leveraging the proposed hard negative contrastive dataset and incorporating triplet contrastive pre-training for compositional data. This approach is orthogonal to prior works.

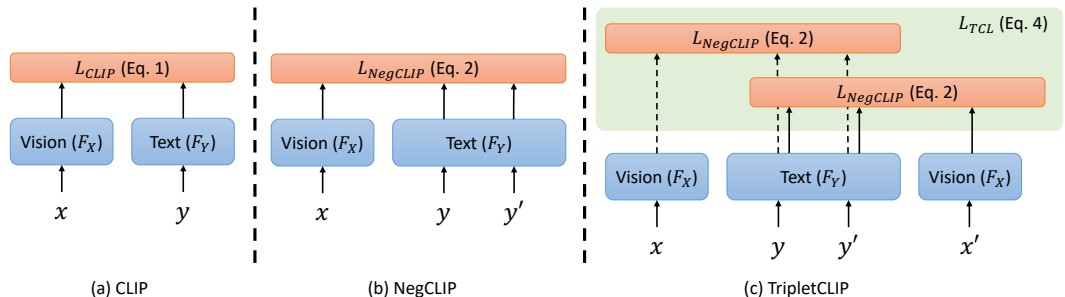

Figure 1: Comparison of training workflows of CLIP, NegCLIP, and `TripletCLIP`. $(x, y)$ represents the positive a image-text pair, and $(x', y')$ represents the corresponding negative image-text pair.

**Data for Contrastive Pre-training.** The effectiveness of maximizing mutual information between modalities heavily relies on the quality of extensive, web-scraped datasets that ideally encompass all possible concepts and knowledge. For instance, despite its noise, the LAION dataset [48, 16], which includes more than 5 billion internet images paired with alt-text captions, is a primary resource. Studies show that over 1 billion data points are necessary to match the performance of the original CLIP model [16, 56]. Recent works like DataComp [16] and MetaCLIP [56] have focused on creating smaller, high-quality datasets by applying stringent filters and ensuring wordnet [33] synset-level concept diversity. Nevertheless, the inherently noisy nature of internet-scraped datasets can degrade model performance. Studies such as SynthCLIP [18] demonstrate that tripling the volume of fully synthetic data is required to equal the efficacy of real data. Other efforts like VeCLIP [25] and LaCLIP [14] enhance dataset quality by using generative language models to re-caption existing images, significantly boosting performance.

**Compositionality for vision-language.** Despite the increased emphasis on data quality and modeling techniques, mastering compositionality remains a significant challenge for vision-language models. Benchmarks like ARO [58], VALSE [35], and CREPE [32] have been developed to assess models' abilities to handle compositional data. SugarCrepe [20], in particular, offers a large-scale, systematic framework for such evaluations. Previous methods primarily focused on identifying hard negatives within existing datasets or generating synthetic negative captions [58, 61, 12, 11, 57, 49]. However, these rule-based generated captions are often unrealistic and linguistically flawed, leading to suboptimal model performance on complex datasets like SugarCrepe. A handful of works focus on finding negative images. [54] propose utilizing the video data. [44] focuses on object-centric image-editing to synthesize the negative images. [39] utilizes the simulation-based data negative data.

Contrary to prior approaches that predominantly add unrealistic negative captions or very constrained negative images that are either very synthetic or object-focused, this work introduces `TripletCLIP`, which centers on generating naturally occurring ***hard negative image-text pairs***. We propose a novel triplet contrastive learning strategy that effectively utilizes these challenging data pairs. Additionally, while our method is distinct, integrating advancements that refine contrastive learning could potentially boost `TripletCLIP`'s efficacy further.

## 3 Method

This section begins with an overview of the contrastive learning algorithm used by CLIP and NegCLIP. We then describe the synthetic data generation pipeline for generating hard negatives using LLMs and T2I models and introduce triplet contrastive learning which forms the basis of `TripletCLIP`. A high-level comparison between prior work and `TripletCLIP` can be found in Figure 1.

### 3.1 Preliminaries

The goal for self-supervised contrastive learning [13], when dealing with inputs from a single modality, is to use a feature extractor ($F$) to encode inputs and their augmentations and minimize

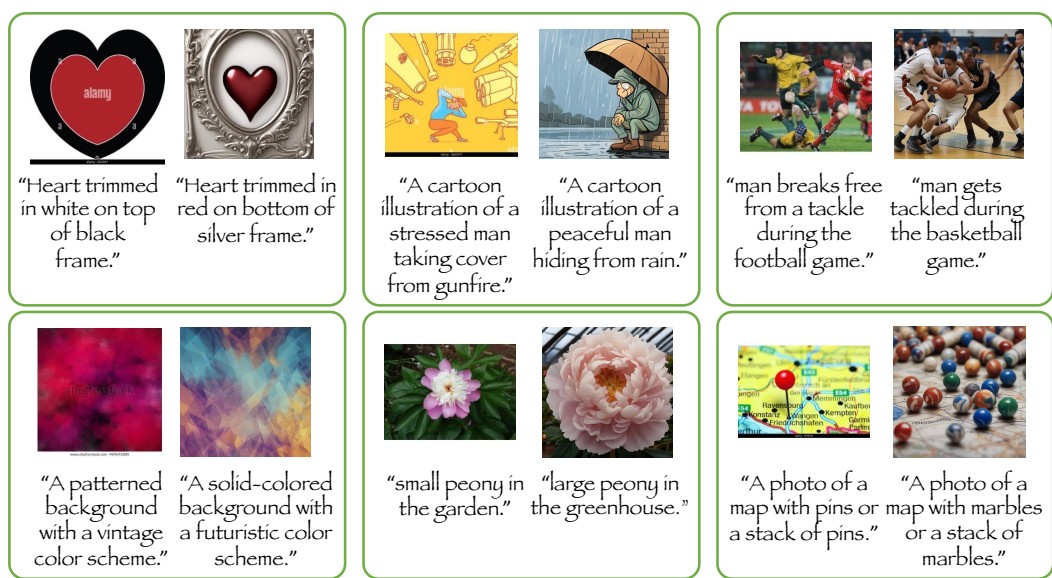

Figure 2: Examples image-text pairs from `TripletData`. In each block, a positive pair from CC3M is on the left and corresponding negatives from `TripletData` are shown on the right.

the InfoNCE loss [34] between the two encodings. CLIP is designed for multimodal settings (for example, vision and language inputs) – this entails using two encoders (one for each modality).

Let $\mathcal{X}$ and $\mathcal{Y}$ represent two modalities and $\mathcal{D} = \{(x_i, y_i)\}_{i=1,...,M}$, be the training dataset, where $x_i \in \mathcal{X}$ and $y_i \in \mathcal{Y}$. The goal is to train two modality-specific encoders, $F_{\mathcal{X}}$ and $F_{\mathcal{Y}}$, by minimizing the InfoNCE loss between the normalized features extracted from the encoders.

$$\mathcal{L}_{\mathcal{X} \to \mathcal{Y}}^{CL} = \frac{-1}{N} \sum_{i=1}^{N} \log \frac{\exp(\langle F_{\mathcal{X}}(x_i), F_{\mathcal{Y}}(y_i)\rangle/\tau)}{\sum_{k=1}^{N} \exp(\langle F_{\mathcal{X}}(x_i), F_{\mathcal{Y}}(y_k)\rangle/\tau)}, \tag{1}$$

where $\langle \cdot \rangle$ represents cosine similarity and $\tau$ is the trainable temperature parameter. For simplicity, we do not show feature normalization in the InfoNCE loss. Similarly, we can define the $\mathcal{L}_{\mathcal{Y} \to \mathcal{X}}^{CL}$ training loss. The combined CLIP total training objective is given as, $\mathcal{L}_{CLIP} = \mathcal{L}_{\mathcal{X} \to \mathcal{Y}}^{CL} + \mathcal{L}_{\mathcal{Y} \to \mathcal{X}}^{CL}$. By minimizing this training loss, both encoders learn representations that maximize the mutual information between two modalities.

NegCLIP introduces synthetic augmentations to generate "hard" negative captions ($y_i' \in \mathcal{Y}'$) by performing semantic inverting perturbations to the reference captions ($y_i \in \mathcal{Y}$). Therefore, the single modality-specific hard negative augmentation-based training loss can be formulated as:

$$\mathcal{L}_{\mathcal{X} \to \mathcal{Y}; \mathcal{Y}'}^{NegCL} = \frac{-1}{N} \sum_{i=1}^{N} \log \frac{\exp(\langle F_{\mathcal{X}}(x_i), F_{\mathcal{Y}}(y_i)\rangle/\tau)}{\sum_{k=1}^{N} \exp(\langle F_{\mathcal{X}}(x_i), F_{\mathcal{Y}}(y_k)\rangle/\tau) + \sum_{m=1}^{N} \exp(\langle F_{\mathcal{X}}(x_i), F_{\mathcal{Y}}(y_m')\rangle/\tau)}. \tag{2}$$

The total loss for NegCLIP for image modality ($\mathcal{X}$) and text modality ($\mathcal{Y}$) is given by:

$$\mathcal{L}_{NegCLIP}(\mathcal{X}, \mathcal{Y}, \mathcal{Y}') = \mathcal{L}_{\mathcal{Y} \to \mathcal{X}}^{CL} + \mathcal{L}_{\mathcal{X} \to \mathcal{Y}; \mathcal{Y}'}^{NegCL}. \tag{3}$$

In Eq. 2, the negative samples are generated only for language modality as it is easy to make semantic-level perturbations. Existing methods have not explored performing semantic perturbations in the image modality to create hard negatives. In this work, we demonstrate how hard negatives can be created in the image modality by leveraging the semantic language grounding and photorealism of text-to-image diffusion models. Our novel hard negative generation pipeline and refined training objective seeks to bridge the significant gap identified in literature.

## 3.2 `TripletData`: Image-text hard negative data augmentations

To generate high-quality hard negative image-text pairs, we follow a two-step procedure. The first stage is to generate hard negative captions from the ground truth positive caption. Second, to generate images corresponding to the hard negative captions as negative images. The AltText captions from the existing web-scrapped datasets are very noisy, leading to the noisy and unreliable generation of hard negatives. Therefore, we build upon the existing work LaCLIP, which first rewrites the captions using LLM from the existing data that are linguistically accurate. Figure 2 illustrates several examples of positive and corresponding negative image-text pairs.

**Generating hard negative captions.** Existing works perform random swapping, replacing, and adding actions between the nouns, attributes, and relations of the positive caption [58, 61]. This method results in nonsensical and grammatically incorrect artifacts, such as "a person riding on four slope," which impedes the generation of negative images, ultimately leading to diminishing performance on harder benchmarks [20]. Therefore, we utilize the in-context learning ability of LLMs to generate negative captions. The choice of LLM is a trivial task as long as they provide hard negative captions. We find that Mistral-7B-Instruct-v0.2[2] performs reasonably better on our goal, and the output is easy to parse. We generate the negative captions in batches to speed up the generation process. Generating the 13M negative captions takes only 3 days on 8xRTX A6000. Instead of generating multiple hard negative captions, we find that a single high-quality hard negative caption is enough to improve the performance compared to the traditional NegCLIP style caption generation (see NegCLIP++ results in Table 4). We provide examples of various types of negative captions in the appendix. Specifically, we provide the following prompt to LLM:

> You are given the description of an image. You should provide a mostly similar description, changing the original one slightly but introducing enough significant differences such that the two descriptions could not possibly be for the same image. Keep the description length the same. Finally, only a few things (such as counting, objects, attributes, and relationships) can be modified to change the image structure significantly. Provide just the updated description. Examples: Input: A dog to the left of the cat. Output: A dog to the right of the cat. Input: A person wearing a red helmet drives a motorbike on a dirt road. Output: A person in a blue helmet rides a motorbike on a gravel path. Now, do the same for the following captions:
> Input: {} Output:

**Generating hard negative images.** Typically, semantic perturbations within images require tools like image editing, which are resource-intensive and cannot be scaled. Remember that we want to provide additional ground truth references for the negative caption. Therefore, we propose to utilize the negative captions from the previous stage to generate the respective reference images directly for pre-training. As the previous stage generates negative captions that are linguistically correct, it becomes easier for image-generative models to synthesize the respective images precisely. We utilize pre-trained text-to-image diffusion models to generate the corresponding images. Specifically, we select SDXL-turbo [46] due to its relatively faster generation speed. After applying various inference time optimizations, we can generate 13M negative images within 2 days using 30 v100 GPUs. We provide various examples of the hard negative image-text pairs in the appendix.

**Analyzing difficulty of the hard `Triplet-Data`.** Let's assume we have positive and negative image-text pairs from the `TripletData`, $(x_i, y_i)$ and $(x_i', y_i')$, respectively. If the data is truly hard negative, existing pre-trained models should struggle to find the correct image-text pairs (i.e., $cos(x_i, y_i) > cos(x_i, y_i')$). Following winoground, we measure the text-score, image-score, and group-score to evaluate the popular pretrained CLIP models. Table 1 shows that even CLIP models trained on billions of data struggle to get near human performance on `TripletData`, which is less difficult than winoground. Importantly, the goal of generating hard negative samples isn't to add more diversity

Table 1: Winoground-style evaluation of pretrained CLIP models on `TripletData`.

|  | Img Score | Text Score | Grp Score |
|---|---|---|---|
| ViT-B/32 | 40.29 | 68.17 | 36.53 |
| ViT-L/14 | 44.84 | 69.21 | 40.91 |
| ViT-bigG | 42.94 | 77.61 | 40.98 |
| Siglip-so400m | 44.24 | 71.27 | 26.10 |
| **Humans (on Winoground)** | **88.50** | **89.50** | **85.50** |

Table 2: Wordnet synset analysis of captions from CC3M and `TripletData`.

|  | CC3M | TripletData (Negative Only) | TripletData | Intersection |
|---|---|---|---|---|
| # unique | 59094 | 59616 | 62741 | **55969** |
| # total synsets | 231M | 215M | 446M | - |

[2]https://huggingface.co/mistralai/Mistral-7B-Instruct-v0.2

Table 3: **Importance of image-text hard negatives.** We measure the importance of various modality-specific hard negatives on SugarCrepe, image-text retrieval, and ImageNet1k. We find that `Triplet-CLIP` results into the most optimal solution. **Bold** number indicates the best performance.

| Models | Negative Captions | Negative Images | SugarCrepe | Retrieval | ImageNet1k |
|---|---|---|---|---|---|
| **LaCLIP** | × | × | 54.09 | 8.19 | 3.79 |
| **NegImage** | × | ✓ | 56.28 | 9.20 | 4.48 |
| **NegCLIP++** | ✓ | × | 61.69 | 8.36 | 3.84 |
| **TripletCLIP** | ✓ | ✓ | **63.49** | **16.42** | **7.31** |

in terms of unique concepts during the training but to add diversity in semantic meanings. Therefore, we measure the unique wordnet synsets in CC3M *vs.* `TripletData`. From Table 2, it can be observed that `TripletData` does not add any new concepts but uses existing concepts to provide negative samples that are semantically different. To summarize, `TripletData` contains the relatively hard negative image-text pairs that current models find difficult to differentiate.

## 3.3 TripletCLIP

Prior works have demonstrated the value of hard negative captions for enhancing the compositionality of CLIP models via $\mathcal{L}_{NegCLIP}$ as the key training objective (Eq. 2) [58, 61]. However, it remains elusive if negative images alone can benefit or not. We conduct modality-specific ablations, reporting the average performance across the diverse set of benchmarks in Table 3 (we provide more details about experiments in Section 4). Our findings indicate that both "hard" negative captions and images individually boost performance when compared to LaCLIP. However, this initial empirical experiments to train the CLIP model on hard negative images (i.e., NegImage) by minimizing $\mathcal{L}_{NegCLIP}(\mathcal{Y}, \mathcal{X}, \mathcal{X}')$ reveal that negative images alone cannot improve the compositionality significantly (see Table 3). We hypothesize that images contain low-level information, making it difficult to train the model using images as negative examples. Aligning with our initial motivation and building upon this crucial insight, we propose to utilize the negative images to regularize the effect of negative captions and to stabilize the pre-training. Therefore, to utilize these hard negative image-text pairs from the previous stage more effectively, we propose to focus on two triplets $(\mathcal{X}, \mathcal{Y}, \mathcal{Y}')$ and $(\mathcal{X}', \mathcal{Y}', \mathcal{Y})$, hence, the final triplet contrastive learning training objective is defined as:

$$\mathcal{L}_{TCL} = \mathcal{L}_{NegCLIP}(\mathcal{X}, \mathcal{Y}, \mathcal{Y}') + \mathcal{L}_{NegCLIP}(\mathcal{X}', \mathcal{Y}', \mathcal{Y}). \tag{4}$$

Intuitively, the second term introduces the additional form of supervision that hard negative images are closer to the corresponding negative captions than positive captions. This allows the system to understand that if the positive image does not represent the negative caption "blue horse," then what does this caption entail? Through this strategic alternation of hard negative image-text pairs for the `TripletCLIP`, we improve compositionality and image-text understanding of the vision-language model (see Table 3). We provide the pseudo-code in the appendix and the code in supplementary materials. This simple yet effective strategy elevates the training of the CLIP, offering a scalable framework to improve overall performance.

## 4 Experiments & Results

### 4.1 Experiment Setup

**Pretraining Datasets.** We utilize the CC3M and CC12M datasets, which comprise 2.6M and 8.6M image-text pairs, respectively. Following the approach demonstrated by LaCLIP, we use LLM-rewritten captions to replace noisy original captions. For NegCLIP, we introduce four negative captions per positive image-text pair, focusing on semantic inverting perturbations across four categories: attribute, relation, object, and action [61]. This generates approximately 10.4M and 34.4M text-only augmentations for CC3M and CC12M, respectively. To train the `TripletCLIP`, we create augmentations (`TripletData`) for both datasets to integrate hard negatives effectively. We produce one augmentation per image-text pair, adding 2.6M and 8.6M image-text augmented pairs for CC3M and CC12M, respectively. Finally, we perform all the ablations on the CC3M dataset.

Table 4: **Composition evaluations of the methods on SugarCrepe benchmark. Bold** number indicates the best performance and underlined number denotes the second-best performance. † represents the results taken from SugarCrepe benchmark.

| Methods | Replace | | | Swap | | Add | | Overall |
|---|---|---|---|---|---|---|---|---|
| | Object | Attribute | Relation | Object | Attribute | Object | Attribute | Avg. |
| **CC3M** LaCLIP | 59.44 | 53.17 | 51.42 | 54.69 | 49.25 | 55.29 | 55.35 | 54.09 |
| LaCLIP + HN | 63.44 | 55.96 | 50.71 | 50.60 | 48.57 | 56.98 | 51.16 | 53.92 |
| NegCLIP | 62.71 | 58.12 | 54.48 | **56.33** | 51.20 | 56.26 | 61.13 | 57.18 |
| NegCLIP++ *(ours)* | 64.77 | 66.12 | **65.93** | 55.51 | 55.41 | 59.65 | **64.45** | 61.69 |
| TripletCLIP *(ours)* | **69.92** | **69.03** | 64.72 | **56.33** | 57.96 | 62.61 | 63.87 | 63.49 |
| **Performance Gain w.r.t. LaCLIP** | 10.48% | 18.56% | 13.30% | 1.64% | 8.71% | 7.32% | 8.52% | 9.40% |
| **CC12M** LaCLIP | 75.06 | 65.48 | 58.68 | 53.47 | 57.66 | 67.65 | 66.76 | 63.54 |
| NegCLIP | 77.84 | 69.29 | 63.23 | **66.53** | 62.31 | 67.17 | 69.65 | 68.00 |
| NegCLIP++ *(ours)* | 82.99 | 78.68 | 75.75 | 61.63 | **65.47** | 70.08 | 76.01 | 72.94 |
| TripletCLIP *(ours)* | **83.66** | **81.22** | **79.02** | 64.49 | 63.66 | **73.67** | 75.43 | **74.45** |
| **Performance Gain w.r.t. LaCLIP** | 8.60% | 15.75% | 20.34% | 11.02% | 6.00% | 8.67% | 7.35% | 10.91% |
| **DataComp** `small`: **ViT-B/32**† (13M) | 56.90 | 56.85 | 51.99 | 50.81 | 50.00 | 53.93 | 60.55 | 54.43 |
| `medium`: **ViT-B/32**† (128M) | 77.00 | 69.54 | 57.68 | 57.72 | 57.06 | 66.73 | 64.88 | 64.37 |
| `large`: **ViT-B/16**† (1B) | **92.68** | 79.82 | 63.94 | 56.10 | 57.66 | **84.34** | 78.61 | 73.31 |
| `xlarge`: **ViT-L/14**† (13B) | **95.52** | **84.52** | 69.99 | **65.04** | 66.82 | **91.03** | 84.97 | **79.70** |

Table 5: **Zero-shot image-text retrieval and classification results. Bold** number indicates the best performance and underlined number denotes the second-best performance.

| Methods | Retrieval (R@5) | | | | Zero-shot Classification | | | |
|---|---|---|---|---|---|---|---|---|
| | Image-to-Text | | Text-to-Image | | VTAB | | ImageNet1k | |
| | MSCOCO | Flickr30k | MSCOCO | Flickr30k | top-1 | top-5 | top-1 | top-5 |
| **CC3M** LaCLIP | 5.06 | 10.90 | 5.97 | 10.84 | 11.56 | 34.72 | 3.79 | 10.49 |
| LaCLIP + HN | 8.08 | 16.10 | 8.64 | 16.64 | **12.31** | 37.14 | 5.75 | 15.22 |
| NegCLIP | 6.32 | 13.80 | 6.61 | 12.96 | 12.25 | 36.38 | 4.67 | 12.69 |
| NegCLIP++ *(ours)* | 5.8 | 11.20 | 6.19 | 10.24 | 11.65 | 35.47 | 3.84 | 10.52 |
| TripletCLIP *(ours)* | **10.38** | **22.00** | **11.28** | **22.00** | **12.31** | **41.45** | **7.32** | **18.34** |
| **Performance Gain** | 5.32% | 11.1% | 5.31% | 11.16% | 0.75% | 6.73% | 3.53% | 7.85% |
| **CC12M** LaCLIP | 25.86 | 42.70 | 19.78 | 36.30 | 19.08 | 49.06 | 19.72 | 41.39 |
| NegCLIP | 30.16 | 46.60 | 23.11 | 41.70 | 19.12 | 50.56 | 20.22 | 42.63 |
| NegCLIP++ *(ours)* | 26.96 | 43.90 | 22.69 | 42.86 | 18.48 | 50.38 | 19.06 | 40.91 |
| TripletCLIP *(ours)* | **33.00** | **55.90** | **28.50** | **52.38** | **20.81** | **53.40** | **23.31** | **47.33** |
| **Performance Gain** | 7.14% | 13.2% | 8.72% | 16.08% | 1.73% | 4.34% | 3.59% | 5.94% |

**Baselines.** We train LaCLIP, LaCLIP with real hard negatives (LaCLIP+HN), and NegCLIP from scratch to ensure consistency and fairness in our comparisons. As NegCLIP's rule-based augmentations closely resemble some compositional benchmarks, so we introduce NegCLIP++ as an improved baseline. NegCLIP++ incorporates hard negative captions generated using LLM from `TripletData`, enhancing the language comprehension compared to standard NegCLIP.

**Implementation Details.** Our experiments employ the ViT-B/32 [10] model architecture. To guarantee fair comparisons, we retrain all baseline models using identical hyperparameters. Since the overall training data for NegCLIP and `TripletData` is more than the baseline datasets, we align the number of iterations across all models to equalize the number of image-text pairs seen during training, similar to the strategy used in DataComp. The batch size is fixed to 1024 with the AdamW optimizer at a maximum learning rate of 0.0005, employing cosine decay. Training durations are set at approximately 100k iterations for CC3M and 200k iterations for CC12M. All models are trained on a single A100 (80GB) GPU using bf16 precision. The final training-related experiments and ablations will cost about 1200 A100 GPU hours. We leave the experiments on increasing the data and model size as future works for the community, as scaling further is not viable in the academic budget.

**Downstream Datasets.** The primary objective of this study is to enhance the compositional capabilities of CLIP models. We mainly evaluate `TripletCLIP` and the baseline models using the challenging SugarCrepe composition benchmark, with additional performance assessments provided in the appendix for older benchmarks. Models are also tested on image-text retrieval tasks for broader evaluation using the Flickr30k [40] and MSCOCO [29] datasets. Zero-shot classification performance

Table 6: **Ablation on filtering high-quality image-text pairs from** `TripletData`. We evaluate the `TripletCLIP` after applying the filters to ensure the quality similar to DataComp and compare the baselines on three benchmarks. We find that `TripletCLIP` results in the most optimal solution. **Bold** number indicates the best performance. † represents that results are borrowed from DataComp.

| Models | Filtering Strategy | Data Size | Augmentations | SugarCrepe | Retrieval | ImageNet1k |
|---|---|---|---|---|---|---|
| **CLIP**[†] | No filtering | 12.8 | - | 55.61 | 6.49 | 2.7 |
| | CLIP Score | 3.8 | - | 57.31 | 9.08 | 5.1 |
| | Image-based ∩ CLIP Score | 1.4 | - | 54.75 | 5.63 | 3.9 |
| **LaCLIP** | No filtering (CC3M) | 2.6 | - | 54.09 | 8.19 | 3.79 |
| **TripletCLIP** | No filtering (CC3M) | 2.6 | 2.6 | 63.49 | 16.42 | 7.31 |
| **TripletCLIP++** | CLIP Score (from CC12M) | 1.4 | 1.4 | **66.09** | **19.85** | **8.85** |

is assessed across approximately 18 different datasets. Evaluations adhere to the methodologies outlined in the CLIP-Benchmark[3] or the official benchmark implementations.

## 4.2 Compositional reasoning

We comprehensively analyze the compositional understanding of models on the SugarCrepe benchmark, as detailed in Table 4. Notably, `TripletCLIP` consistently outperforms all baseline models across all sub-categories of SugarCrepe on both the CC12M/CC3M training datasets. Specifically, `TripletCLIP` surpasses LaCLIP and NegCLIP by **10.91%/9.4%** and **6.45%/6.31%** on the CC12M and CC3M datasets, respectively. Our enhanced baseline, NegCLIP++, also shows improvement over standard NegCLIP, highlighting the benefits of LLM-generated negatives. Nevertheless, `Triplet-CLIP` further advances performance, underscoring the critical role of hard negative image-text pairs, not just text. Additional comparisons on older composition benchmarks (Valse [35], Cola [42], and Winoground [53]) in the appendix reveal `TripletCLIP`'s consistent performance. Table 4 also contrasts `TripletCLIP` with models trained using the DataComp approach, which involves more parameters and training data, demonstrating that `TripletCLIP` achieves comparable performance to a ViT-B/16 model trained on 1 billion image-text pairs.

## 4.3 Zero-shot evaluations

**Image-Text Retrieval.** In Table 5, we summarize the performance of models on text-to-image (T2I) and image-to-text (I2T) retrieval tasks on MSCOCO and Flickr30k datasets, where we report R@5 scores. Remarkably, `TripletCLIP` significantly outperforms baseline models by an average of **8%/10%** and **8%/12.5%** on I2T and T2I tasks, respectively, on the CC3M and CC12M datasets. Intriguingly, while LaCLIP+HN performs better than NegCLIP, `TripletCLIP` outstrips both.

**Zero-shot Classification.** Table 5 also presents the average zero-shot classification performance on 18 standard datasets, including ImageNet1k. `TripletCLIP` consistently enhances top-1 accuracy by an average of **3%** and top-5 accuracy by **5-7%** compared to LaCLIP. Like the retrieval performance, LaCLIP+HN exceeds NegCLIP, yet `TripletCLIP` maintains the highest performance. Dataset-specific results are in the appendix.

## 4.4 Finetuning performance

In this paper, we focus on pretraining-based experiments as they allow greater flexibility in learning better representations. To complement this, we also performed additional fine-tuning experiments using hyperparameters similar to the baselines (without LoRA) and compared them against various publicly available baselines [44, 50, 12, 11]. As reported in Table 7, `TripletCLIP` improves compositionality and outperforms nearly all baselines. Furthermore, the observed drop in retrieval and zero-shot classification performance (Table 16) is attributed to limitations in the vision encoder, highlighting the challenges of existing pre-trained vision encoders in capturing semantic representations. This is further demonstrated in Table 8.

---

[3] https://github.com/LAION-AI/CLIP_benchmark

Table 7: **Finetuning-based composition evaluations of the methods on SugarCrepe benchmark.** **Bold** number indicates the best performance and underlined number denotes the second-best performance.

| Methods | Replace | | | Swap | | Add | | Overall |
|---|---|---|---|---|---|---|---|---|
| | Object | Attribute | Relation | Object | Attribute | Object | Attribute | Avg. |
| CLIP | 90.92 | 80.08 | 69.13 | 61.22 | 64.26 | 77.16 | 68.64 | 73.06 |
| CLIP (finetuned) | 90.92 | 79.69 | 64.01 | 60.82 | 64.26 | 84.67 | 78.76 | 74.73 |
| NegCLIP | 91.53 | 83.25 | 73.97 | 72.24 | 67.72 | 86.95 | 88.44 | 80.59 |
| Baseline [44] | 93.22 | 84.39 | 67.35 | 62.04 | 70.12 | 88.31 | 79.48 | 77.84 |
| CoN-CLIP [50] | 93.58 | 80.96 | 63.3 | **87.29** | **79.62** | 59.18 | 65.16 | 75.58 |
| TSVLC (RB) [12] | 91.34 | 81.34 | 64.15 | 68.16 | 69.07 | 79.49 | 91.33 | 77.84 |
| TSVLC (LLM+RB) [12] | 88.13 | 76.78 | 62.73 | 64.08 | 66.67 | 75.80 | 81.07 | 73.61 |
| DAC [11] | **94.43** | **89.48** | **84.35** | 75.10 | 74.17 | 89.67 | **97.69** | **86.41** |
| TripletCLIP *(ours)* | **94.43** | 85.53 | 80.94 | 69.80 | 69.82 | **90.40** | 86.27 | 82.46 |

Table 8: **Frozen encoder ablation.** LiT style fine-tuning ablations on SugarCrepe, image-text retrieval, and ImageNet1k. **Bold** number indicates the best performance.

| Models | Train Text | Train Vision | SugarCrepe | Retrieval | ImageNet1k |
|---|---|---|---|---|---|
| LaCLIP | ✓ | ✗ | **0.6373** | 0.5345 | 31.21% |
| TripletCLIP *(ours)* | ✓ | ✗ | 0.6227 | **0.6817** | **34.25%** |
| LaCLIP | ✗ | ✓ | 0.5886 | 0.1134 | 5.51% |
| TripletCLIP *(ours)* | ✗ | ✓ | **0.6923** | **0.2626** | **12.51%** |

## 4.5 Ablations

**Can a high-quality filtered dataset improve the performance?** Given that negative images in `TripletData` are generated using SDXL-turbo, these may not always be precise. Inspired by Data-Comp, we employ a pre-trained CLIP-L/14 to filter the image-text pairs, selecting the highest average similarity pairs (positive and negative) individually (i.e., score = $\left(s(x_i, y_i) + s(x'_i, y'_i)\right)/2$). The top 1.4M positive image-text pairs and their corresponding negatives from `TripletData` are selected. Table 6 details this comparison against DataComp pre-trained models. Remarkably, `TripletCLIP` already surpasses baselines without filtered data; however, with the filtered dataset, despite being trained on 50% smaller dataset, `TripletCLIP++` shows further performance improvements. This underlines the significant benefits of carefully selected `TripletData` in enhancing the performance.

**Which modality-specific encoder plays the key role in improving compositionality?** To address this open question, we designed an ablation study similar to LiT, freezing either the pre-trained CLIP vision or text encoder while training the opposite modality-specific encoder from scratch. We observe the performance of LaCLIP and `TripletCLIP` on CC3M, as shown in Table 8. Freezing the vision model results in no performance gain on the SugarCrepe for `TripletCLIP`. However, significant improvements are noted when the vision encoder is actively trained, suggesting that the vision modality may be the bottleneck in compositionality. Notably, `TripletCLIP` outperforms LaCLIP in all settings, further demonstrating its robustness to different pre-training approaches.

**Concept coverage analysis.** Improving performance on zero-shot transfer learning tasks such as retrieval involves two key components: adding more concept diversity during training and enhancing image-text alignment/compositionality. We create subsets of CC12M data with increasing concept diversity based on unique WordNet synsets. Specifically, we select 3M, 4M, 5M, and 6M subsets for training LaCLIP, while `TripletCLIP` training involves only half of these training data as positive pairs, and the rest are corresponding augmentations. Evaluations across SugarCrepe, retrieval tasks, and ImageNet1k (see Figure 3) indicate that `TripletCLIP` not only enhances SugarCrepe performance even at lower concept coverage levels but also significantly outperforms similar concept coverage in retrieval tasks, matching LaCLIP's performance on zero-shot classification tasks that do not require compositionality at all. This bolsters our argument that incorporating hard negatives from both modalities markedly improves compositional understanding in CLIP, while baseline struggles to do so even with more concept diversity.

**What if `TripletData` is used for large-scale compositional evaluations?** We evaluated the CC12M pre-trained models on a 50,000 random subset of the CC3M dataset using a Winoground-style

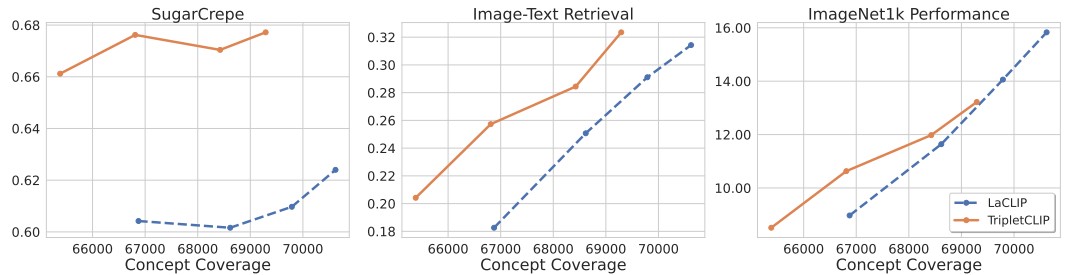

Figure 3: Average Results of LaCLIP and `TripletCLIP` for SugarCrepe Compositions, Image-Text Retrieval, and ImageNet1k over increasing concept diversity.

Table 9: TripletData as large-scale composition evaluation dataset after [53].

| Methods | Text-Score | Image-Score | Group-Score |
|---|---|---|---|
| **CLIP** | 52.69 | 29.66 | 24.64 |
| **NegCLIP** | 54.84 | 30.42 | 25.82 |
| **NegCLIP++** | 36.50 | 30.67 | 20.11 |
| **TripletCLIP** *(ours)* | **92.25** | **66.82** | **64.30** |

approach [53]. As shown in Table 9, `TripletCLIP` significantly improves performance compared to the baselines. However, we partially attribute this improvement to spurious correlations learned from the data. At the same time, we note that the models have not fully converged, suggesting minimal risk of overfitting to these spurious correlations.

## 5 Conclusion

In this work, we introduce `TripletCLIP`, a novel approach to enhancing compositional reasoning in vision-language models through the strategic incorporation of hard negative image-text pairs. Our comprehensive experiments across a suite of benchmarks demonstrate that `TripletCLIP` significantly outperforms existing methodologies such as LaCLIP and NegCLIP, achieving notable gains not only in compositionality but also in zero-shot classification and retrieval tasks as well. Further, our ablation studies highlight the critical role of modality-specific training and the careful curation of training data, underscoring the importance of both hard negative image and text components in the learning process. `TripletCLIP`'s effectiveness with a smaller, refined dataset suggests a promising direction for future research—maximizing performance without the need for extensive data collection, thereby reducing computational costs and enhancing model efficiency. To this end, we provide an intriguing application of synthetic datasets via hard negative image-text pairs for vision-language tasks that could be easily extended to improve Multimodal Large Language Models and Text-to-Image generative models.

**Limitations.** Due to constraints inherent in academic settings and limited computational resources, we were unable to scale `TripletCLIP` to handle hundreds of millions of image-text pairs or employ larger models within the scope of this study. Nevertheless, our results indicate a promising direction for future research within a consistent experimental framework, and we encourage subsequent work to explore scaling both the `TripletData` and `TripletCLIP`. Our experimental focus was primarily on the CLIP and LiT methodologies. With additional resources, however, extending our methodologies to more advanced contrastive learning techniques, such as SigLIP, would be feasible. In conclusion, our work introduces a compelling strategy for integrating open-ended hard negatives (both text and image) during the pre-training phase, providing a methodology and large-scale data that could benefit a variety of research domains.

## Acknowledgments

This work was supported by NSF RI grants #1750082, #2132724, and CPS grant #2038666. We thank the Research Computing (RC) at Arizona State University (ASU) for providing computing resources. The views and opinions of the authors expressed herein do not necessarily state or reflect those of the funding agencies and employers.

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

## A   Broader Impact

In this study, we have demonstrated the potential of utilizing high-quality positive and negative pairs to enhance the compositional understanding of vision-language models like CLIP through the introduction of `TripletCLIP`. While our findings are specific to `TripletCLIP`, the underlying techniques hold promise for broader applications, including enhancing visual understanding in Multimodal Large Language Models (MLLMs) and Text-to-Image diffusion models. Although our approach yields significant performance improvements, it does require resources to generate large-scale synthetic datasets. We encourage future research to explore the utility of this pre-training strategy within the latent space, which could reduce dependence on large generative models. Initiatives like JEPA [1] have already demonstrated the efficacy of focusing on latent space models, which suggests a promising avenue for reducing computational overhead. Importantly, our experiments reveal that significant enhancements in model performance are achievable even with substantially smaller data scales. This finding suggests that, when scaled appropriately, our methodology could substantially diminish resource dependencies and enhance the efficiency of pre-training processes for CLIP-like models.

## B   Pseudocode of TripletCLIP

```
def forward(batch_positive, batch_negative):
    # get positive and negative image-text pairs
    img_pos, txt_pos = batch_positive
    img_neg, txt_neg = batch_negative

    # compute positive image and text representations
    image_features_pos = clip.encode_image(img_pos)
    text_features_pos = clip.encode_text(txt_pos)

    # compute negative image and text representations
    image_features_neg = clip.encode_image(img_neg)
    text_features_neg = clip.encode_text(txt_neg)

    positive_txt = torch.cat([text_features_pos, text_features_neg])
    negative_txt = torch.cat([text_features_neg, text_features_pos])

    # compute NegCLIP losses
    loss_1 = negclip_loss(image_features_pos, positive_txt)
    loss_2 = negclip_loss(image_features_neg, negative_txt)
    loss = loss_1 + loss_2

    return loss
```

## C   Hyperparameters

Table 10: Detailed pre-training hyper-parameters for CLIP training across various experiments and ablations.

| Hyperparameters | CC3M | CC12M | LiT | Concept Coverage Ablations |
|---|---|---|---|---|
| Batch size | 1024 | 1024 | 1024 | 1024 |
| Optimizer | AdamW | AdamW | AdamW | AdamW |
| Learning rate | $5 \times 10^{-4}$ | $5 \times 10^{-4}$ | $5 \times 10^{-4}$ | $5 \times 10^{-4}$ |
| Weight decay | 0.5 | 0.5 | 0.5 | 0.5 |
| Adam $\beta$ | (0.9, 0.999) | (0.9, 0.999) | (0.9, 0.999) | (0.9, 0.999) |
| Adam $\epsilon$ | $1 \times 10^{-8}$ | $1 \times 10^{-8}$ | $1 \times 10^{-8}$ | $1 \times 10^{-8}$ |
| Total steps | 90,000 | 230,000 | 90,000 | 200,000 |
| Learning rate schedule | cosine decay | cosine decay | cosine decay | cosine decay |

Table 10 provides a comprehensive overview of the pre-training hyperparameters employed across all baseline models and `TripletCLIP`. To ensure fair comparisons, we standardized the hyperparame-

Table 11: **Composition evaluations of the methods on various benchmarks.** **Bold** number indicates the best performance and underlined number denotes the second-best performance.

| | Methods | Valse | Cola | | | Winoground | | | SugarCrepe | Overall |
|---|---|---|---|---|---|---|---|---|---|---|
| | | | Txt2Img | Img2Txt | Group | Txt2Img | Img2Txt | Group | | |
| CC3M | LaCLIP | 43.19 | 28.10 | 13.33 | 10.48 | **24.75** | 9.25 | 6.00 | 54.09 | 23.65 |
| | LaCLIP + HN | 47.91 | 20.95 | 5.54 | 2.38 | 23.25 | 4.25 | 3.00 | 53.92 | 20.15 |
| | NegCLIP | 48.06 | 21.42 | 14.76 | 8.57 | 21.00 | 8.50 | 5.25 | 57.18 | 23.09 |
| | NegCLIP++ *(ours)* | 43.65 | 29.05 | 13.33 | 7.14 | 22.00 | 5.50 | 3.25 | 61.69 | 23.20 |
| | TripletCLIP *(ours)* | 48.36 | 31.43 | 13.33 | 9.52 | 24.25 | 6.25 | 4.25 | 63.49 | 25.11 |
| | TripletCLIP++ *(ours)* | 48.53 | 31.43 | 15.71 | 12.86 | 22.25 | 9.50 | 6.75 | 66.09 | 26.64 |
| | **Performance Gain *w.r.t.* LaCLIP** | **5.34%** | **3.33%** | **2.38%** | **2.38%** | **-2.50** | **0.25%** | **0.75%** | **12.00** | **2.99%** |
| CC12M | LaCLIP | 55.69 | 20.95 | 15.71 | 7.62 | **26.25** | **8.00** | **6.25** | 67.21 | 25.96 |
| | NegCLIP | 58.59 | 27.14 | 15.24 | 5.71 | 18.25 | 6.50 | 4.25 | 68.41 | 25.51 |
| | NegCLIP++ *(ours)* | 58.12 | 33.33 | 11.43 | 7.14 | 23.50 | 7.75 | 5.50 | 73.05 | 27.48 |
| | TripletCLIP *(ours)* | 57.57 | 27.62 | 19.53 | 11.43 | 23.25 | 6.25 | 4.25 | 74.55 | 28.06 |
| | **Performance Gain *w.r.t.* LaCLIP** | **1.88%** | **6.67%** | **3.82%** | **3.81%** | **-3.00** | **-1.75** | **-2.00** | **7.35** | **2.10%** |

Table 12: **Dataset-specific zero-shot classification results.** **Bold** number indicates the best performance and underlined number denotes the second-best performance.

| | Models | fgvc aircraft | imagenet1k | caltech101 | cifar100 | clevr closest object distance | clevr count all | dmlab | dsprites label orientation | dsprites label x position | dtd | eurosat | flowers | kitti closest vehicle distance | pcam | pets | smallnorb label azimuth | smallnorb label elevation | svhn | Avg. |
|---|---|---|---|---|---|---|---|---|---|---|---|---|---|---|---|---|---|---|---|---|
| CC3M | LaCLIP | 0.84 | 3.79 | 24.65 | 8.04 | 15.86 | 11.26 | **19.02** | **2.63** | 3.14 | 5.53 | 8.56 | 3.43 | 26.72 | 48.16 | 3.11 | 5.75 | 10.86 | 6.68 | 11.56 |
| | LaCLIP+HN | 1.23 | 5.75 | 33.20 | 8.84 | 17.18 | 11.18 | 16.60 | 2.52 | 3.10 | 6.65 | 10.02 | 4.29 | 18.85 | 51.93 | 4.44 | 4.92 | 11.12 | 9.78 | 12.31 |
| | NegCLIP | 1.02 | 4.67 | 29.44 | 8.17 | 22.37 | 12.97 | 18.81 | 2.57 | 3.19 | 6.81 | 7.39 | 3.77 | 28.83 | 41.72 | 4.17 | 4.84 | 11.37 | 8.32 | 12.25 |
| | NegCLIP++ *(ours)* | 1.02 | 3.84 | 20.61 | 6.44 | 22.03 | 10.01 | 18.91 | 2.60 | 3.21 | 4.63 | 11.70 | 3.59 | 23.91 | 49.34 | 4.74 | 5.43 | 10.02 | 7.70 | 11.65 |
| | TripletCLIP *(ours)* | **1.29** | 7.32 | 37.05 | **14.37** | 15.55 | 13.49 | 15.37 | 2.62 | 2.98 | 8.78 | 22.74 | 5.97 | 16.74 | 53.49 | 5.81 | 5.34 | 11.51 | **10.22** | 12.31 |
| | TripletCLIP++ *(ours)* | 0.90 | 8.85 | 38.54 | 11.83 | 20.72 | 13.79 | 17.46 | 2.52 | 3.58 | 8.99 | 24.43 | 8.20 | 19.41 | 56.60 | 19.38 | 5.76 | 12.46 | 7.91 | 15.62 |
| CC12M | LaCLIP | **1.59** | 19.72 | 62.98 | 23.62 | 11.99 | 11.38 | **19.85** | 2.91 | **3.20** | 13.03 | **23.85** | 13.43 | 24.05 | 50.38 | **34.72** | **6.14** | 11.04 | 9.54 | 19.08 |
| | NegCLIP | 1.56 | 20.22 | 63.45 | 26.02 | 18.89 | 15.80 | 16.20 | 2.45 | 3.13 | 14.26 | 17.43 | 13.55 | 22.78 | 50.22 | 30.85 | 5.37 | 12.04 | 10.04 | 19.12 |
| | NegCLIP++ *(ours)* | 1.38 | 19.06 | 64.66 | 24.86 | 15.96 | 14.87 | 15.62 | 2.94 | 3.07 | 13.62 | 14.33 | 14.30 | 16.60 | 49.98 | 31.92 | 5.42 | 12.08 | 12.13 | 18.48 |
| | TripletCLIP *(ours)* | 1.29 | 23.31 | 65.34 | 30.30 | 16.26 | 17.67 | 16.58 | 2.80 | 3.18 | 17.45 | 23.26 | 15.43 | 25.74 | 51.04 | 33.25 | 5.58 | 12.38 | 13.67 | 20.81 |

ters across all methodologies. Although larger batch sizes are typically associated with improved performance in contrastive learning, computational constraints necessitated fixing the batch size at 1024 for all experiments. To accommodate this batch size on a single A100 GPU, we employed bf16 precision. In terms of computational resources, experiments using the CC3M dataset required approximately 16 GPU hours, while those involving the CC12M dataset utilized up to 56 GPU hours per experiment.

# D    Detailed Results

## D.1    Compositional reasoning

Previously, we reported the results on SugarCrepe, the most challenging dataset, noted for its absence of language biases. However, evaluations were also conducted on other benchmarks, such as Valse, Cola, and Winoground. As indicated in Table 11, `TripletCLIP` achieves overall improvements of **2-3%** compared to LaCLIP and NegCLIP. The Valse benchmark, which contains text prompts that heavily favor the perturbations made for NegCLIP, shows a strong performance from NegCLIP, while NegCLIP++ encounters difficulties. Interestingly, `TripletCLIP` faces challenges in maintaining performance on Winoground, and baseline LaCLIP maintains the SOTA, which is counterintuitive to other benchmarks. Nonetheless, `TripletCLIP` still manages to outperform NegCLIP significantly. These results affirm that `TripletCLIP` sets a new standard for state-of-the-art compositional reasoning across diverse benchmarks.

## D.2    Dataset-specific zero-shot classification

Table 12 provides fine-grained results for the 18 zero-shot classification datasets. It can be observed that `TripletCLIP` consistently outperforms the baselines, achieving the best average results across these challenging datasets. Although the improvements are marginal, they are in line with expectations. As discussed in Figure 3, `TripletCLIP` does not introduce new concepts into the training data but focuses on augmentations that enhance representation without increasing concept diversity. These

Table 13: **Composition evaluations of the methods on various benchmarks. Bold** number indicates the best performance and underlined number denotes the second-best performance.

| | Methods | Text-to-Image Retrieval | | | | | | Image-to-Text Retrieval | | | | | |
|---|---|---|---|---|---|---|---|---|---|---|---|---|---|
| | | MSCOCO | | | Flickr30k | | | MSCOCO | | | Flickr30k | | |
| | | R@1 | R@5 | R@10 | R@1 | R@5 | R@10 | R@1 | R@5 | R@10 | R@1 | R@5 | R@10 |
| CC3M | LaCLIP | 1.56 | 5.06 | 8.90 | 3.70 | 10.90 | 16.00 | 1.73 | 5.97 | 9.50 | 3.54 | 10.84 | 16.02 |
| | LaCLIP + HN | 2.60 | 8.08 | 13.02 | 6.30 | 16.10 | 22.40 | 2.66 | 8.64 | 13.38 | 5.98 | 16.64 | 23.82 |
| | NegCLIP | 1.74 | 6.32 | 10.44 | 4.90 | 13.80 | 19.60 | 1.95 | 6.61 | 10.62 | 4.76 | 12.96 | 18.50 |
| | NegCLIP* | 1.50 | 5.80 | 9.70 | 4.40 | 11.20 | 16.30 | 1.85 | 6.19 | 9.82 | 3.50 | 10.24 | 15.20 |
| | TripletCLIP | **3.16** | **10.38** | **16.22** | **9.10** | **22.00** | **29.80** | **3.58** | **11.28** | **17.39** | **8.38** | **22.00** | **29.56** |
| | **Performance Gain vs. CLIP** | 1.60% | 5.32% | 7.32% | 5.40% | 11.10% | 13.80% | 1.85% | 5.31% | 7.89% | 4.84% | 11.16% | 13.54% |
| CC12M | LaCLIP | 10.50 | 25.86 | 35.60 | 21.30 | 42.70 | 54.60 | 7.21 | 19.78 | 28.37 | 15.06 | 36.30 | 47.00 |
| | NegCLIP | 12.32 | 30.16 | 41.44 | 24.70 | 46.60 | 58.20 | 8.56 | 23.11 | 32.60 | 18.66 | 41.70 | 53.42 |
| | NegCLIP* | 10.94 | 26.96 | 36.64 | 18.70 | 43.90 | 55.90 | 8.80 | 22.69 | 32.13 | 18.24 | 42.86 | 53.78 |
| | TripletCLIP | **14.60** | **33.00** | **43.84** | **28.00** | **55.90** | **65.70** | **11.38** | **28.50** | **39.04** | **25.28** | **52.38** | **63.32** |
| | **Performance Gain vs. CLIP** | 4.10% | 7.14% | 8.24% | 6.70% | 13.20% | 11.10% | 4.17% | 8.72% | 10.67% | 10.22% | 16.08% | 16.32% |

Table 14: **Ablation on choice pre-trained LLM.** We train NegCLIP++ (ViT-B/32) on negative captions generated from various LLMs and report SugarCrepe, Flickr30k Retrieval (R@5), and ImageNet-top5 performances.

| Models | SugarCrepe (avg.) | Retrieval (R@5) | ImageNet1k (top-5) |
|---|---|---|---|
| **Gemma-2b-it** | 56.00 | 12.60 | **12.09%** |
| **Phi-3-mini-4k-instruct** | 61.22 | **13.02** | 10.94% |
| **Mistral-7b-instruct-v0.2** | **61.69** | 10.72 | 10.52% |

Table 15: **Ablation on CyCLIP with TripletLoss.** To evaluate the compatibility of TripletLoss with the CyCLIP, we train the ViT-B/32 models from search on CC3M with 512 batch size and report SugarCrepe, Flickr30k Retrieval (R@5), and ImageNet-top5 performances.

| Models | SugarCrepe (avg.) | Retrieval (R@5) | ImageNet1k (top-5) |
|---|---|---|---|
| LaCLIP | 55.11 | 12.80 | 12.58% |
| **TripletCLIP** | **65.71** | **24.62** | **19.95%** |
| CyCLIP | 54.62 | 11.77 | 13.01% |
| **CyCLIP+TripletLoss** | **58.64** | **20.29** | **19.05%** |

enhanced representations from `TripletCLIP` lead to average improvements of **1-3%** depending on the scenario.

## D.3  Detailed image-text retrieval performance

Table 13 presents detailed T2I and I2T retrieval results for the MSCOCO and Flickr30k datasets. We report results at different recall thresholds: R@1, R@5, and R@10. The data shows that `TripletCLIP` significantly outperforms the baselines across all recall rates. On average, `Triplet-CLIP` achieves a performance gain of **7-11%** over LaCLIP. Additionally, `TripletCLIP` improves performance by **3-6%** compared to previous state-of-the-art baselines.

## D.4  Additional ablations

**Choice of Pre-trained LLM.**   We provide further details regarding the selection of LLMs for generating hard negative captions. NegCLIP++ was trained on 3 million generated negative captions for CC3M using three different LLMs, and the results are reported in Table 14. We find that Phi-3 achieves the best average performance, while Gemma-2b unexpectedly has a notable negative impact on compositionality. However, we opted to use Mistral-7b-instruct-v0.2, as Phi-3 was released after our experiments were completed, preventing its earlier evaluation.

**Evaluating the Orthogonality of TripletLoss.**   As discussed in Section 2, `TripletCLIP` can be integrated with various previously proposed methodologies. To evaluate this, we applied TripletLoss in conjunction with CyCLIP [17] and reported the results in Table 15. The most significant performance improvement is observed with the TripletLoss alone. However, TripletLoss also enhances the performance over the base CyCLIP, demonstrating its adaptability and orthogonality with respect to other approaches.

Table 16: **Finetuning-based evaluations of the methods on Retrieval and ImageNet-1k benchmarks. Bold** number indicates the best performance and underlined number denotes the second-best performance.

| Methods | Retrieval | | | | ImageNet 1k | |
| | Text Retrieval (R@5) | | Image Retrieval (R@5) | | | |
| | MSCOCO | Flickr30k | MSCOCO | Flickr30k | top-1 | top-5 |
|---|---|---|---|---|---|---|
| **CLIP** | **74.9** | 94.60 | 55.92 | 83.38 | **63.31** | **88.22** |
| **CLIP (finetuned)** | 68.9 | 88.40 | 53.50 | 81.10 | 49.95 | 79.16 |
| **NegCLIP** | 66.00 | 88.60 | 53.41 | 81.12 | 48.85 | 78.34 |
| **Baseline [44]** | 81.4 | **96.0** | **67.49** | **89.84** | 61.40 | 88.10 |
| **TSVLC (RB) [12]** | 71.70 | 93.00 | 62.01 | 87.12 | 58.81 | 85.97 |
| **TSVLC (LLM+RB) [12]** | 71.82 | 92.50 | 62.24 | 87.46 | 59.77 | 87.02 |
| **DAC [11]** | 54.5 | 79.60 | 63.51 | 87.84 | 51.02 | 81.22 |
| **TripletCLIP** *(ours)* | 55.6 | 82.60 | 53.32 | 80.88 | 45.92 | 75.54 |

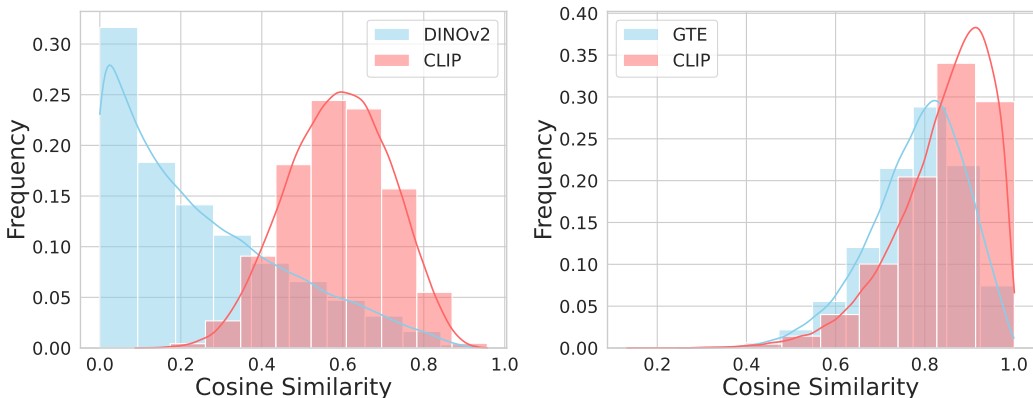

Figure 4: Positive *vs.* Negative modality-specific pair-based similarity distribution of pre-trained CLIP ViT-B/32 model *w.r.t.* the vision and text-only encoders. The left plot is the vision embedding similarities between positive and negative images. The right plot is the text embedding similarities between positive and negative captions. In the ideal scenario, the distribution should be skewed towards 0.0, which indicates that the model can correctly distinguish between the positive and negative data.

# E   Encoder Representation Distribution Analysis

Remember, this study aims to learn the representations that can distinguish between two data points that are very similar but semantically different. Firstly, we take LaCLIP and `TripletCLIP` models trained on CC12M. We also sampled 50000 positive+negative pairs from CC3M. Then, we measure the vision and text modality-specific cosine similarities between positive and negative pairs and plot the distribution (see Figure 5). It can be observed that vision representations from `TripletCLIP` are more skewed towards 0.0, suggesting that the vision encoder can distinguish between hard negative samples better than the baseline LaCLIP. However, in the case of the text modality, both methods perform similarly. This aligns with our findings from Table 8 that the vision encoder plays a crucial role in improving the compositionality, and to achieve this, our `TripletData` is necessary.

# F   `TripletData` **Analysis**

This section provides qualitative examples of the `TripletData` and discusses various data analyses.

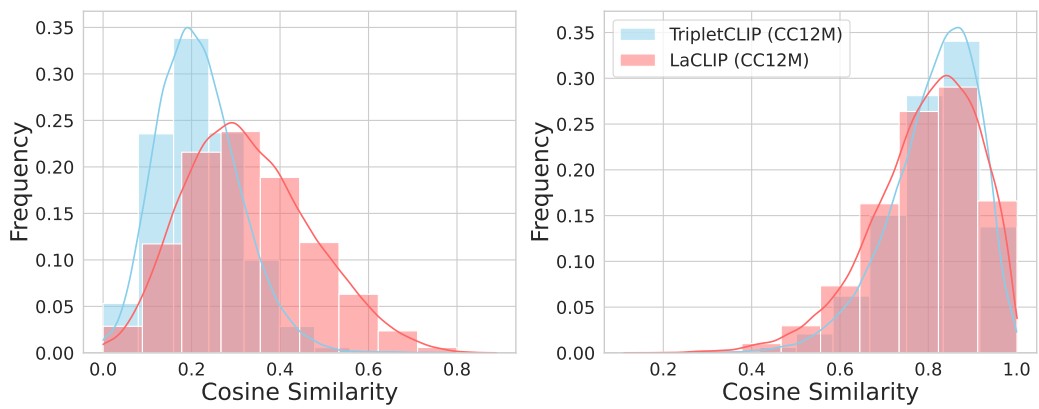

Figure 5: Positive *vs.* Negative modality-specific pair-based similarity distribution of baseline LaCLIP and `TripletCLIP`. The left plot is the vision embedding similarities between positive and negative images. The right plot is the text embedding similarities between positive and negative captions.

Table 17: **Question Generation.** Examples of LLM-generated existence-related questions from captions to evaluate the generated images.

| Captions | Questions |
|---|---|
| A smooth, flat sheet of woven fabric (which looks like woven silk) is shown in closeup. | Is the entity a sheet? 
 Is the sheet made of fabric? 
 Is the fabric woven? 
 Does the fabric look like silk? 
 Is the fabric flat and smooth? |
| The sunset over a calm sea. | Is there a sunset? 
 Is the sea calm? 
 Is the sunset over the sea? 
 Is the sunset happening during the day? 
 Is the sea rough? |
| the businesswoman finished second and descended the podium of the runners-up. | Did the businesswoman finish second? 
 Did the businesswoman descend the podium? 
 Are there runners-up? 
 Is the podium for the second place? 
 Is the businesswoman a runner? |
| Kid standing still near a scooter in a public place. | Is there a kid in the public place? 
 Is the kid near a scooter? 
 Is the scooter in a public place? 
 Is the kid standing still? 
 Is there a public place in the caption? |
| A local resident drives in the fall on a scenic drive, wearing a red scarf. | Is the local resident driving? 
 Is it happening in the fall? 
 Is there a scenic drive? 
 Is the local resident wearing a red scarf? 
 Is the local resident walking? |

**Difficulty of the data.** We further add one more analysis to investigate how difficult our dataset is. First, we take state-of-the-art language-only (GTE [28]) and vision-only (DINO [4]) embedding models and pretrained CLIP ViT-B/32. Later, we measure the modality-specific similarity between positive and negative vision and language data pairs. Figure 4 shows that the similarity distribution of the pretrained CLIP model between positive and negative text pairs follows the distribution of the text-only GTE model. Interestingly, vision distribution is drastically different. DINO can distinguish the positive and negative pairs correctly with high confidence. However, despite the visuals being so different, the pretrained CLIP model struggles to distinguish the different images. This further highlights that `TripletData` is indeed challenging for the vision-language models, even the ones trained on large-scale datasets.

**Evaluations of Generated Images.** Even though T2I diffusion models are widely evaluated on various tasks. We perform additional evaluations to measure how accurately generated images follow

the text prompt. To do this, taking inspiration from [36], we first use LLM to generate binary "yes/no" style questions from the given caption. As shown in Table 17, we create five questions per hard negative caption. Later, we utilized the ViLT model to answer visual questions. Upon this investigation, we find that SDXL-turbo archives on an average of 76% accuracy. In other words, the T2I model can correctly generate an image that follows around 3/4th of the text. Additionally, we hypothesize that using an improved T2I model or image editing models to generate "hard" negative examples can further improve composition reasoning.

**Qualitative Examples.** In Figure 6, we provide additional qualitative examples of the contrastive positive and "hard" negative pairs from the `TripletData`. Additionally, in Figure 7, we illustrated several examples where the T2I model could not precisely generate images corresponding to the caption. However, we may notice that in most cases, it maintains some of the important aspects. Because of this, despite not being 100% accurate all the time, it can help `TripletCLIP` improve performance across the evaluation benchmarks.

**Hard negative caption only examples:**

1. **Raw Caption:** dog looking out from a window .
   **Language Rewrite:** A dog looking through the window at his owner.
   **Negative Caption (NegCLIP):**
   (a) A window looking through the dog at his owner.
   (b) A dog looking through the window at his dog.
   (c) A dog screams through the window at his owner.
   `TripletData` **Negative Caption:** A cat observing its owner from the window.
2. **Raw Caption:** person attends the premiere of film
   **Language Rewrite:** A person attends the premiere of film
   **Negative Caption (NegCLIP):**
   (a) A premiere attends the person of film
   (b) A person attends the festival of film
   (c) A person watches the premiere of film
   `TripletData` **Negative Caption:** A person waits in line for film tickets.
3. **Raw Caption:** white crocus spring flowers in the forest.
   **Language Rewrite:** white crocus flowers in the forest
   **Negative Caption (NegCLIP):**
   (a) white crocus flowers in the sky
   `TripletData` **Negative Caption:** red orchid flowers in the meadow.
4. **Raw Caption:** flag with industry in the background
   **Language Rewrite:** A flag is holding in the background an industrial site.
   **Negative Caption (NegCLIP):**
   (a) A background is holding in the flag an industrial site.
   (b) A flag is holding in the background an earthquake site.
   (c) A drone is holding in the background an industrial site.
   (d) A flag is visible in the background an industrial site.
   `TripletData` **Negative Caption:** A flag is flapping in the foreground of a pastoral scene.
5. **Raw Caption:** portrait of businessman with cardboard on his head carrying a briefcase and using an umbrella while standing by.
   **Language Rewrite:** A portrait of a businessman standing by with a briefcase and cardboard on his head carrying an umbrella while looking at a blue sky and parked cars on the street
   **Negative Caption (NegCLIP):**
   (a) A businessman of a portrait standing by with a briefcase and cardboard on his head carrying an umbrella while looking at a blue sky and parked cars on the street
   (b) A portrait of a businessman standing by with a briefcase and cardboard on his head carrying an umbrella while looking at a grey sky and parked cars on the street
   (c) A portrait of a businessman standing by with a briefcase and cardboard on his head carrying an umbrella while looking at a blue truck and parked cars on the street
   (d) A portrait of a businessman standing by with a briefcase and cardboard on his head carrying an umbrella while pointing at a blue sky and parked cars on the street
   `TripletData` **Negative Caption:** A portrait of a businesswoman seated on a bench with a tote bag and newspaper on her lap holding an umbrella while looking at a red sunset over row houses.

6. **Raw Caption:** 158834 is the portion of the bound train .
   **Language Rewrite:** Huge locomotives sit on the tracks in front of a building.
   **Negative Caption (NegCLIP):**
   (a) Huge tracks sit on the locomotives in front of a building.
   (b) Three locomotives sit on the tracks in front of a building.
   (c) Huge locomotives sit on the tracks in anticipation of a building.
   (d) Huge locomotives mounted on the tracks in front of a building.
   `TripletData` **Negative Caption:** Huge locomotives sit on the tracks in front of a bridge.
7. **Raw Caption:** biological subfamily eating fish on a seaweed covered shore
   **Language Rewrite:** Two blue whales are eating salmon on a beach surrounded by seaweed
   **Negative Caption (NegCLIP):**
   (a) Two blue salmon are eating whales on a beach surrounded by seaweed
   (b) Two stranded whales are eating salmon on a beach surrounded by seaweed
   (c) Two blue whales are eating salmon on a farm surrounded by seaweed
   (d) Two blue whales are eating salmon on a beach covered by seaweed
   `TripletData` **Negative Caption:** Two blue whales are feeding on herring in a bay surrounded by kelp.
8. **Raw Caption:** image of an original oil painting on canvas
   **Language Rewrite:** A young lady holding a painting to your face so you can see the detail of the painting
   **Negative Caption (NegCLIP):**
   (a) A young painting holding a lady to your face so you can see the detail of the lady
   (b) A bearded lady holding a painting to your face so you can see the detail of the painting
   (c) A young lady holding a pencil to your face so you can see the detail of the painting
   (d) A young lady holding a painting to your face so you can enjoy the detail of the painting
   `TripletData` **Negative Caption:** A young lady holding a painting away from her face to show its beauty to the audience.

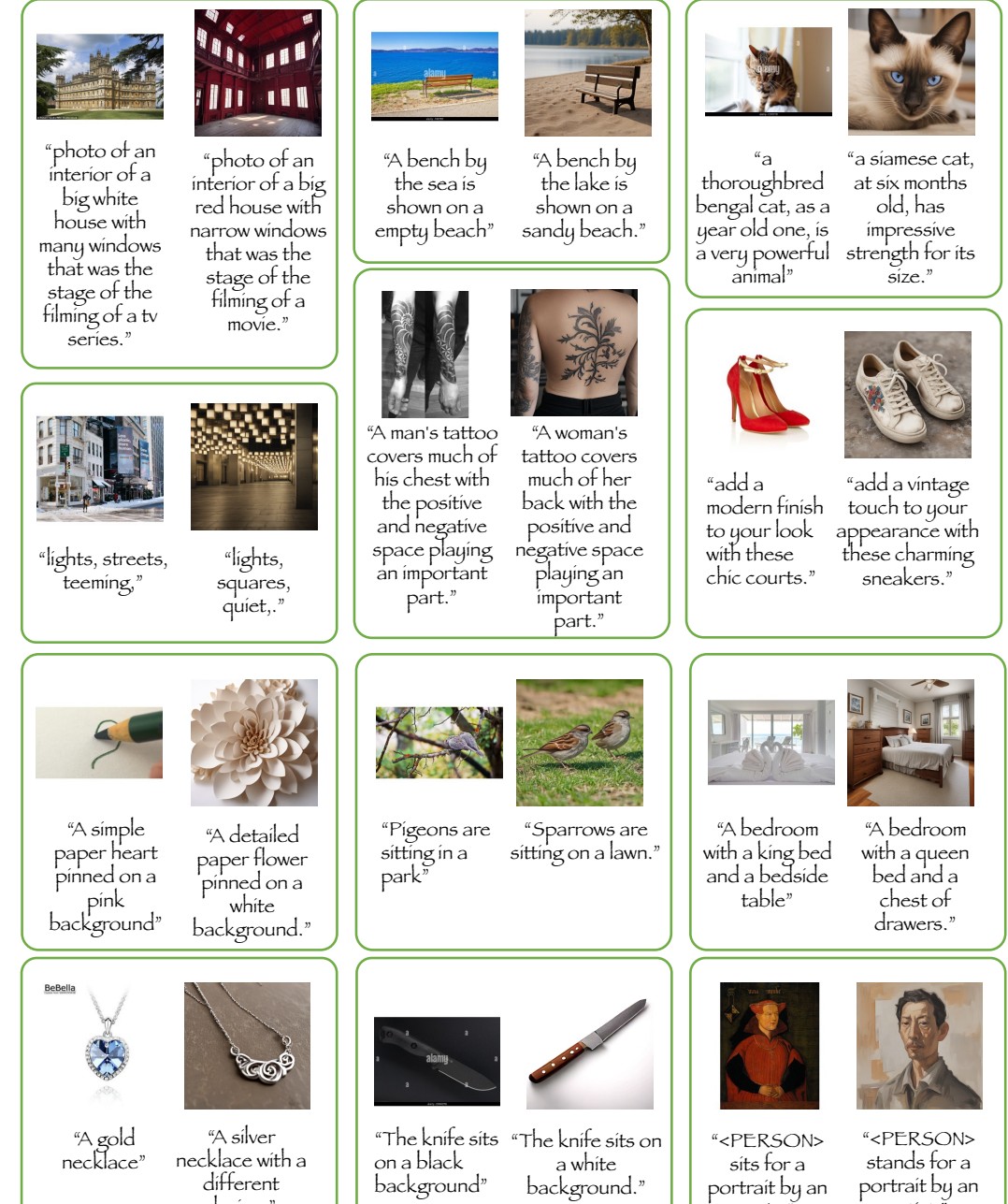

Figure 6: Qualitative examples of positive and hard negative image-text pairs from `TripletData`. In each block, left image-text pairs are positive images from CC3M, and right pairs are corresponding negatives from `TripletData`.

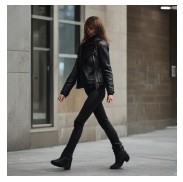 "someone entered a building looking cool in a black leather jacket as she walked."

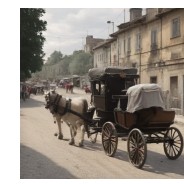 "A small car at the side of the road where a traditional horse-drawn carriage is driven by a woman."

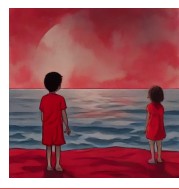 "Standing next to a red ocean, two children gaze up at the half moon with a promise to one day swim across the seas"

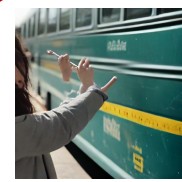 "A woman is using a ruler to measure the distance between two points outside of a bus."

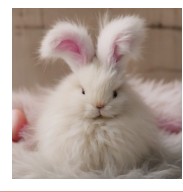 "A decorative Easter egg is balanced precariously on a fluffy bunny's head."

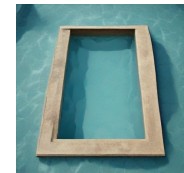 "An hourglass in the shape of a rectangular pool."

Figure 7: Examples of T2I failures.

