# OpenReview forum: "TripletCLIP:  Improving Compositional Reasoning of CLIP via Synthetic Vision-Language Negatives"
_NeurIPS.cc/2024/Conference — NeurIPS 2024 poster_

### Official Review · Reviewer_m3ja · 2024-06-30

**Soundness:** 4
**Presentation:** 3
**Contribution:** 4
**Rating:** 7
**Confidence:** 3

**Summary:**

The paper proposes enhancing CLIP's compositional reasoning by generating high-quality negative image-text pairs using LLMs and text-to-image models. This approach improves beyond previous works that considered unrealistic, rule-based captions and unexplored negative images. Experiments show consistent improvements across different tasks.

**Strengths:**

- **Dataset**: The paper proposes using LLMs and text-to-image models to generate hard negative captions and images, respectively. The generation and filtering methodology is effective. I find that releasing TripletData alone is already valuable for the community.
- **Comprehensive experiments**: The paper shows extensive evaluation and consistent improvements over baselines on compositional and downstream tasks like SugarCrepe, zero-shot classification, and image-text retrieval.
- **Clear presentation**: The paper is well-structured and clearly written, explaining the methodology and results effectively.

**Weaknesses:**

- **Limited scaling**: The authors acknowledged the lack of experiments on increasing data and model size due to academic budget constraints.
- **Limited study on the selection of LLM and image-to-text model**: The authors acknowledged the limited study on how the selection of the models could affect the quality of the TripletData.
- **Nit**: Consider using \log and \exp in the equations.

**Questions:**

- Have you considered splitting the test set and evaluating the models on the TripletData?
- Could you clarify how the total of 13M image-text pairs is calculated? The numbers provided (2.6M for CC3M and 8.6M for CC12M) in Section 4.1 add up to 11.2M.
- Since the negative images are generated from a text-to-image model, there's likely a distribution mismatch from the real positive images. Have you inspected this potential discrepancy?
- Can you elaborate on the relatively poor performance on the Winoground benchmark compared to other compositional tasks? Are there specific aspects of Winoground that TripletCLIP struggles with?
- There's a mention of "MiraData" in the Appendix - is this a typo for TripletData, or is it referring to something else?

**Limitations:**

The authors have clearly addressed limitations in their paper.

---

> ### Author Rebuttal · Authors · 2024-08-06
>
> We are encouraged by your review! We thank you for your comprehensive evaluation of our paper. We are grateful that you found TripletData alone to be of significant value and that our experiments are comprehensive.
>
> Please find the requested clarifications below.
>
> > ## Response to Weaknesses
>
> **[W1] Limited Scaling:** We acknowledge that scaling our experiments further in academic settings is challenging. However, to approximate this, we experimented with increasing the data (i.e., WordNet synsets), as shown in Figure 3. Our findings indicate a consistent upward trend, with TripletCLIP outperforming the baseline CLIP in terms of compositionality while maintaining zero-shot classification performance.
>
> **[W2] Selection of LLMs and T2I Models:** Grounding the selection procedure of LLM to generate negative captions is indeed essential. We performed additional experiments with three LLMs and trained NegCLIP++ to evaluate their behaviors. The results show that Phi-3 is the best overall. However, at the time of submission, Phi-3 wasn’t released, so we selected the current second-best option, Mistral. The selection of T2I models is straightforward, as their goal is to generate images that truthfully align with the text. Therefore, selecting the best open-source T2I model is a logical choice.
>
> | Models                      |    SugarCrepe    |    Retrieval (R@5)    |    ImageNet1k (top-5)    |
> |-----------------------------|:----------------:|:---------------------:|:-----------------------:|
> | **Gemma-2b-it**                 |       56.00      |        _0.1260_       |        **12.09%**       |
> | **Phi-3-mini-4k-instruct**      |      _61.22_     |       **0.1302**      |         _10.94%_        |
> | **Mistral-7b-instruct-v0.2**    |      **61.69**   |        0.1072         |          10.52%         |
>
>
> **[W3]** We will update the equations in the final draft (if accepted). Thank you!
>
>
> > ## Response to Questions
>
> **[Q1] Evaluations on valdiation subset of TripletData:** Thank you for suggesting this interesting analysis. We evaluated the CC12M pre-trained models on a 50,000 random subset of the CC3M dataset and reported the scores below. As expected, TripletCLIP significantly boosts the performance of the baselines. However, we also partially attribute this to the spurious correlation learned from the data (related to Q3).
> At the same time, we note that models are not fully converged, so there is very little chance of overfitting on this spurious correlation.
>
> | Model           |    Text Score    |    Image Score    |    Group Score    |
> |-----------------|:----------------:|:-----------------:|:-----------------:|
> | **CLIP**            |      52.69       |       29.66       |       24.64       |
> | **NegCLIP**         |      _54.84_       |      30.42      |      _25.82_      |
> | **NegCLIP++**       |      36.50        |      _30.67_    |       20.11       |
> | **TripletCLIP (ours)**     |     **92.25**    |      **66.82**    |      **64.30**     |
>
> **[Q2] Clarification on TripletData size:** We created TripletData on top of LaCLIP, which provides a ~2.9M subset of captions for CC3M and a ~10M subset for CC12M, totaling ~12.9M. However, we observed failures in downloading several images during training, leading to a further decrease in image-text pairs during the training phase. We will clarify this in the final draft (if accepted).
>
> **[Q3] Potential real vs. synthetic distribution mismatch:** Yes, a distribution mismatch exists between real and synthetic images. The generated images are of higher quality than CC3M/12M, potentially adding spurious correlations (partially observed in the Q1 table). However, other generated images represent similar objects with different semantics/relations that are not present in the real data. Despite TripletCLIP learning these biases, it still improves compositionality.
>
> **[Q4] Winoground clarification:** Winoground data contains only 400 instances, which may not be statistically significant for evaluating models trained with lower resources. Even with finetuning on top of the pretrained models, we observed a decrease in performance. Ideally, TripletData should replace the Winoground. We will plan to release the high-quality subset of TripletData as an evaluation set for the community to rely on. Thank you for inspiring this idea.
>
> | Models              |    Text-score    |    Image-score    |    Group-score    |
> |---------------------|:----------------:|:-----------------:|:-----------------:|
> | **CLIP (org)**          |    **0.3125**    |      _0.1100_     |    **0.0875**     |
> | **CLIP (finetuned)**    |      _0.2975_    |      0.0875       |      0.0625       |
> | **NegCLIP**             |      0.2700      |      0.0875       |      _0.0700_       |
> | **TripletCLIP**         |      0.2700      |    **0.1125**     |      _0.0700_       |
>
>
> **[Q5]** Yes, this is a typo. We meant TripletData. Thank you for pointing this out.
>
> We hope this clarifies all the remaining questions. Additionally, we offer a summary of responses to other reviews for your reference in the global response.

---

> > ### Comment · Reviewer_m3ja · 2024-08-11
> >
> > Thank you for the detailed response. I appreciate additional experiments that you provided. I have no further concerns. Overall, I’m looking forward to the dataset that you will provide.

---

> > > ### Author Response · Authors · 2024-08-12
> > > **Reply to Reviewer m3ja**
> > >
> > > Thank you for all your time and attention given to our work. We are pleased to see that all concerns have been effectively addressed.
> > >
> > > Yes, we are planning to release the all versions of the TripletData (CC3M, CC12M, and High-quality filtered version -- 2.8M).

---

### Official Review · Reviewer_f4Gf · 2024-07-03

**Soundness:** 2
**Presentation:** 3
**Contribution:** 2
**Rating:** 6
**Confidence:** 4

**Summary:**

This paper proposes a TripletCLIP, a pre-training framework aimed at enhancing compositional reasoning task. It computes the NegCLIP objective separately for each, with the image serving as the anchor in each instance. For training TripletCLIP, such triplet data is constructed as follows: Hard negative captions are generated from the original LaCLIP captions using a large language model (LLM) through in-context learning mechanism. Hard negative images are created from text-to-image diffusion models. The experiment section compared various hard negative (HN) based objectives and demonstrated the superior performance of the proposed TripletCLIP mechanism across a wide range of benchmarks: compositionality, zero-shot retrieval, and classification.

**Strengths:**

[Originality and Significance]
This paper addresses both model and data aspects in the context of pre-training CLIP. Given the challenges of synthesizing hard negative texts and particularly images at scale, the proposed work for constructing triplet data is likely to be valuable to the research community. On the model side, the proposed hard negative objective in pre-training, though simple, consistently enhances performance across a variety of tasks, including compositionality, retrieval, and classification.

[Quality and Clarity]
The experiments are well-designed and meticulously verify the claims made in the paper, demonstrating the necessity of such design choices. Some additional baselines, beyond the default LaCLIP, are included to demonstrate the effectiveness of TripletCLIP. The overall presentation was clear.

**Weaknesses:**

- [W1] The effectiveness of the TripletCLIP objective is less convincing. Although the paper considered some hard negative (HN)-based baselines such as NegImage and NegCLIP++, it did not compare with HN baselines that jointly consider both HN text and image in the loss calculation, such as [1, 2, 3, 4, 5]. I understand that some works are concurrent, but it would be more comprehensive to see whether the design of the TripletCLIP objective is optimal, either quantitatively or conceptually. In addition, it is necessary to note the 'previous' work that generates hard negative images to enhance compositional reasoning tasks [2, 3]. Accordingly, the sentence from [L136-137] needs to be adjusted.


- [W2] Though there are consistent improvements, the absolute level of performance is relatively weak compared to the 'default' CLIP models from the CyCLIP [1] and ALIP [6] papers (in terms of zero-shot imagenet and retrieval tasks). This may be due to the reduced computational resources during pre-training, which limits the significance of the results in the context of pre-training. The only possibility left, and neither the triplet data nor the pre-trained model shows the evidence of benefiting further downstream tasks in applications such as advanced VLMs like LLaVA, or text-to-image generation, as noted in [L70-72].


- [W3] Connected to [W2], if only a single GPU is used for pre-training, it would be an attractive alternative to fine-tune the pre-trained CLIP model with the (subset of) triplet data. It's clear that the proposed methodology remains valid, and it would be interesting to see whether employing triplet data with the proposed hard negative objectives proves superior compared to other compositional reasoning methodologies in the context of fine-tuning [7, 8], specifically on both compositional reasoning and retrieval/classification tasks. If the authors wish to focus solely on the pre-training context, referencing other previous work discussing both pre-training, as well as fine-tuning is missed [9].


- [W4] It is suggested to consider including additional compositional benchmarks for evaluation that consist of counterfactual image-text pairs, similar to the Winoground style: EqBen [2], COCO-Counterfactuals [10], MMVP-VLM [11], and SPEC [4], for comprehensiveness.


In summary, the main concerns are the missing comparisons with hard negative (HN) objectives that incorporate both image and text, and the limited significance due to small-scale pre-training. Fine-tuning could be considered as an alternative approach to address these issues. Superiority over previous fine-tuning methodologies can further increase the significance. Meanwhile, I believe the large-scale triplet data constructed in the paper is valuable and could promote further studies in various aspects.

---

References

[1] Goel et al., CYCLIP: Cyclic Contrastive Language-Image Pretraining, in NIPS 2022.

[2] Wang et al., Equivariant Similarity for Vision-Language Foundation Models, in ICCV 2023.

[3] Sahin et al., Enhancing Multimodal Compositional Reasoning of Visual Language Models with Generative Negative Mining, in WACV 2024.

[4] Peng et al., Synthesize, Diagnose, and Optimize: Towards Fine-Grained Vision-Language Understanding, in CVPR 2024.

[5] Singh et al., Learn “No” to Say “Yes” Better: Improving Vision-Language Models via Negations, in arXiv preprint 2024.

[6] Yang et al., ALIP: Adaptive Language-Image Pre-training with Synthetic Caption, in ICCV 2023.

[7] Doveh et al., Teaching Structured Vision&Language Concepts to Vision&Language Models, in CVPR 2023.

[8] Doveh et al., Dense and Aligned Captions (DAC) Promote Compositional Reasoning in VL Models, NIPS 2023.

[9] Singh et al., Coarse-to-Fine Contrastive Learning in Image-Text-Graph Space for Improved Vision-Language Compositionality, in EMNLP 2023.

[10] Le et al., COCO-Counterfactuals: Automatically Constructed Counterfactual Examples for Image-Text Pairs, in NIPS Dataset and Benchmark Track, 2023.

[11] Tong et al., Eyes Wide Shut? Exploring the Visual Shortcomings of Multimodal LLMs, in CVPR 2024.

**Questions:**

- Missing citations [L17-19]; image classification [CITATION], segmentation [CITATION], and image-text retrieval [CITATION]

- From [L180], it needs referring to the proper section in the appendix

- From [L227], what is the meaning of the 'real' hard negatives?

**Limitations:**

Limitations are adequately and honestly addressed in the appendix.

---

> ### Author Rebuttal · Authors · 2024-08-06
>
> We appreciate your review of TripletCLIP and the time you've dedicated to it. Thank you for the detailed and holistic review of our work. We are delighted that you found it valuable to the research community.
>
> > ## Comparison with Related Works [W1]
>
> Firstly, we want to clarify the key baselines selected in the main paper, reflecting the current state of research at the pretraining stage:
>
> - NegCLIP: SOTA baseline for negative captions based on the latest CVPR’24 work [12].
> - NegCLIP++: LLM-enhanced negative caption similar to [6,7].
> - CLIP + HN: Real-world hard negatives filtered from existing datasets.
>
> We further divided the related works into two categories:
>
> ### Orthogonal Works [1,6-9]
>
> Our contributions are orthogonal, meaning our findings can be combined with theirs for further improvements. For instance, CyCLIP [1] introduces a novel regularizer but does not include hard negatives. [6,7] introduce synthetic captions similar to NegCLIP++. [8] focuses on detailed caption-based NegCLIP fine-tuning. [9] introduces scene graph-based negative captions.
>
> We retrained CLIP and CyCLIP models with and without TripletLoss using the CC3M training dataset and a batch size of 512. **Adding TripletLoss consistently improved performance over baseline CLIP and CyCLIP**:
>
> | Models | SugarCrepe | Retrieval (R@5) | ImageNet1k (top-5) |
> |--------|:-----:|:------:|:--------:|
> | CLIP | 55.11| 0.1280| 12.58% |
> | TripletCLIP| 65.71| 0.2462| 19.95% |
> | CyCLIP | 54.62| 0.1177| 13.01% |
> | CyCLIP+TripletLoss | 58.64| 0.2029| 19.05% |
>
> ### Negative Captions and Images [2-5]
>
> These works are closer to TripletCLIP. Specifically, [2] utilizes the video dataset to perform hard negative mining and [3] focuses on object-centric caption and image editing for hard negatives, while we focus on free-form captions and image editing that modifies semantics globally. [4] introduces a synthetic data pipeline for benchmarking and proposes a simple extension to NegCLIP pretraining, **aligning with our analysis in Table 3, which shows TripletCLIP’s significance**. [5] focuses on a "negation" and utilizes a method similar to [4].
>
> Although we could incorporate these negative captions [6-9] along with the TripletCLIP procedure for further improvements, due to time constraints, we perform direct comparisons with finetuning-based experiments and leave the joint optimization as future work.
>
> We will include this detailed comparisons in the related work to make it more comprehensive in the final draft.
>
> > ## Finetuning Based Experiments [W2 + W3]
>
> Our paper focuses on pre-training, as fine-tuning focuses on post-training solutions, which may not be generalizable. That said, we performed additional finetuning experiments with hyper-parameters similar to [3,5,7] (w/o LoRA) and compared against various baselines [3,5,7,8], _whose public checkpoints are available_. **Our results show that TripletCLIP improves compositionality** and outperforms the almost all baselines. Additionally, the drop in retrieval and zero-shot classification is attributed to the vision encoder (Table 7), indicating the limitations of existing pre-trained vision encoders to represent semantics. This another reason to conduct the pretraining based experiments for comprehensive evaluations.
>
> | Models | SugarCrepe | Retrieval (R@5) | ImageNet1k (top-5) |
> |------|:-----:|:----:|:-------:|
> | CLIP (org) | 73.06 | 0.8899| 88.82% |
> | CLIP (finetuned) | 74.73| 0.8475| 79.16% |
> | NegCLIP| 80.59| 0.8486| 78.34% |
> | Baseline [3] | 77.84| 0.9292| 88.10% |
> | CoN-CLIP [5] | 75.58| -| -|
> | TSVLC (RB) [7]| 77.84| 0.9006| 85.97% |
> | TSVLC (LLM + RB) [7] | 73.61  | 0.8998| 87.02% |
> | DAC [8]| 86.41| 0.8372| 81.22% |
> | **TripletCLIP (ours)**| 82.46| 0.8174| 75.54% |
>
> Here, DAC [8] introduces detailed human-annotated captions with additional loss functions to incorporate multiple sentences of the captions, which is crucial this performance. Our work focuses on synthetic data with short captions. Future work could explore combining both approaches together.
>
> > ## Additional Clarifications on W2
>
> Lines 70-72 convey that CLIP models are key components behind VLMs and T2I models. Our method can potentially improve CLIP models, directly impacting downstream tasks. TripletData could inspire future improvements (noted by ZTUH), possibly requiring different pretraining like DPO. We leave this as future work. However, we will make it clear in the camera-ready draft.
>
> > ## Additional Benchmark Evaluations
>
> We have **already provided additional comparisons on three widely adopted benchmarks in Table 9**. Importantly, the newly suggested benchmarks [2,4,10,11] are not well-adopted by the community. But below are the additional evaluations on COCO-counterfactuals [10] that is very similar to [2]. Interestingly, all models perform consistently with no significant improvements over the baseline, possibly due to the evaluation data's difficulty or noise. Having said that, due to the time constraints associated with rebuttal phase, we could not extend large-scale evaluations on another two benchmarks.
>
> | Model  | Text-Score | Image-Score | Group-Score |
> |------|:-----:|:-----:|:----:|
> | CLIP| 26.1 | 24.77 | 16.73 |
> | NegCLIP| 26.06| 25.55 | 17.56 |
> | NegCLIP++| 27.21| 25.6  | 16.82 |
> | TripletCLIP  | 26.47| 25.37 | 17.92 |
>
> > ## Response to Remaining Minor Questions
>
> - [Q1] We will add the suggested citations in the final draft (if accepted).
> - [Q2] We will clarify the section we are referring to, specifically Figure 2 (main paper) and Figure 6 (appendix).
> - [Q3] By “real” hard negatives, we mean the hard negative image-text mining within the training datasets instead of synthesizing them.
>
> We trust that our response adequately addresses your concerns and encourages you to reevaluate our submission. We look forward to the discussion.
>
> ---
>
> [12] Zhang et. al., “Contrasting Intra-Modal and Ranking Cross-Modal Hard Negatives to Enhance Visio-Linguistic Compositional Understanding,” CVPR 2024.

---

> ### Comment · Reviewer_f4Gf · 2024-08-09
>
> Thank you for the detailed response. I have no remaining concerns and will adjust the score to 6.
>
> I believe that including additional experiments with CyCLIP and fine-tuning comparisons will provide valuable references for future research.
> It would be great if that part of the codebase were released.

---

> > ### Author Response · Authors · 2024-08-12
> > **Reply to Reviewer f4Gf**
> >
> > Thank you for raising the score! We are pleased to see that all concerns have been effectively addressed.
> >
> > Yes, we will add all the new experiments in the camera-ready version as they will make our work more impactful. We are also planning to release the code, data, and checkpoints.

---

### Official Review · Reviewer_U1Yz · 2024-07-11

**Soundness:** 2
**Presentation:** 2
**Contribution:** 2
**Rating:** 5
**Confidence:** 4

**Summary:**

To enhance the compositional capabilities of CLIP, authors propose to generate “hard” negative captions via in-context learning and synthesizing corresponding negative images with text-to-image generators offers a solution.

**Strengths:**

1. Authors introduce a novel CLIP pre-training strategy that employs hard negative images in conjunction with triplet contrastive learning to enhance compositionality.
2. TripletCLIP consistently improves across downstream tasks.
3. A new dataset TripletData is proposed.

**Weaknesses:**

The novelty and contribution are limited. While [1] builds negative samples from the text perspective, similar to [1], authors primarily build additional negative samples from the image perspective with LLM and generated images.

Given the existing observations (bag-of-words phenomenon, etc.) and soulutions presented in [1], I think the contribution of the current version is marginal.


### Reference
[1] When and why vision-language models behave like bags-of-words, and what to do about it? ICLR 2023

**Questions:**

see weakness

---

> ### Author Rebuttal · Authors · 2024-08-06
>
> Thank you for your review. We respectfully disagree with the claims regarding our work's limited novelty and contributions. Other reviewers have unanimously recognized numerous strengths in our work despite its straightforward nature. Therefore, we would like to reiterate our research's key strengths and contributions.
>
>
> > ## Novelty and Contributions
>
> **Importance:** CLIP models have demonstrated limited performance in terms of compositionality and tend to behave as bags-of-words, as observed by [1]. This work suggested introducing negative mining, particularly negative captions, could enhance model performance. Following this, several studies have focused on improving synthetic hard negative captions [2,3,4,5], while others have aimed at enhancing data quality [6,7,8]. However, with recent advances in image generative models, it remains unclear whether such tools can improve compositionality. Another recent study introduced training CLIP on a fully synthetic dataset [9], but it required three times more data to achieve performance comparable to real data.
>
> **Novelty/Contributions:** To address these limitations and improve image-text understanding, we propose TripletCLIP. TripletCLIP leverages LLMs to generate hard negative captions (similar to [3,4]) and introduces hard negative images focusing on various image semantics. Additionally, we introduce a novel contrastive pretraining loss, TripletLoss, which enhances the usability of our synthetic data generation pipeline. To be specific, our contributions are as follows:
>
> - **TripletCLIP significantly improves baseline performance by an absolute 6-10%** across benchmarks and shows absolute 4-7% improvements over the baseline proposed by [1].
> - **We have released approximately 13M synthetic image-text pairs**, complementing real-world datasets like CC3M and CC12M.
> - **Additional experiments (Table 7) indicate that vision encoders are the primary source of compositionality limitations** in pretrained CLIP models, which were previously unknown. TripletCLIP offers a promising solution to overcome this through careful training on hard negative images, specifically using TripletLoss.
> - **Experiments with increasing concept diversity (Figure 3) further validate our approach**, demonstrating that TripletCLIP consistently improves performance with larger dataset sizes.
>
> We trust that our response adequately addresses your concerns regarding novelty and encourages you to reevaluate our submission. We also kindly request you to consider the feedback from other reviewers and our responses, contributing to a comprehensive assessment of our work.
>
> ---
>
> [1] When and why vision-language models behave like bags-of-words, and what to do about it? ICLR 2023.
>
> [2] Contrasting Intra-Modal and Ranking Cross-Modal Hard Negatives to Enhance Visio-Linguistic Fine-grained Understanding, CVPR 2024.
>
> [3] Dense and Aligned Captions (DAC) Promote Compositional Reasoning in VL Models, NeurIPS 2023.
>
> [4] Teaching Structured Vision&Language Concepts to Vision&Language Models, CVPR 2023.
>
> [5] Coarse-to-Fine Contrastive Learning in Image-Text-Graph Space for Improved Vision-Language Compositionality, EMNLP 2023.
>
> [6] DataComp: In search of the next generation of multimodal datasets, NeurIPS 2023.
>
> [7] Demystifying CLIP Data, ICLR 2024.
>
> [8] DreamLIP: Language-Image Pre-training with Long Captions, ArXiv 2024.
>
> [9] SynthCLIP: Are We Ready for a Fully Synthetic CLIP Training?, ArXiv 2024.

---

> > ### Comment · Reviewer_U1Yz · 2024-08-11
> >
> > Thanks for authors' explanation. Although the methodology is simple and the results are actually not surprising to me as it is pretty intuitive that constructing such “hard” negative would lead to improvements, the additional analytical experiments and analysis could be interesting. I would raise my score to 5.

---

> > > ### Author Response · Authors · 2024-08-12
> > > **Reply to Reviewer U1Yz**
> > >
> > > Thank you for raising the score. While incorporating hard negative scores might seem obvious, much of the existing literature overlooks the significance of negative images and how to utilize them effectively. Through this work, we aim to offer a more comprehensive perspective on compositionality and demonstrate how recent generative models can aid in this, even in their simplest forms.

---

### Official Review · Reviewer_ZTUH · 2024-07-13

**Soundness:** 3
**Presentation:** 3
**Contribution:** 2
**Rating:** 6
**Confidence:** 3

**Summary:**

The paper introduces a novel pre-training strategy aimed at enhancing the compositional reasoning capabilities of CLIP models. The authors identify the limitation in current image-text datasets that restricts the compositional understanding of CLIP models and propose a solution that involves generating "hard" negative captions and corresponding images. This is achieved through a two-step process: leveraging in-context learning for negative caption generation and utilizing text-to-image generators to create matching negative images. The improvement in modeling effectiveness is more significant with the manufactured negative sample dataset.

**Strengths:**

[S1] This paper is well-written, making it easy for readers to follow.
[S2] The method of the paper is concise and easy to implement, and the experimental results demonstrate its effectiveness, which could encourage more people to apply this method to their own tasks.
[S3] For multimodal contrastive learning pre-training, incorporating both image negative samples and text negative samples is comprehensive.

**Weaknesses:**

This research primarily involves leveraging LLMs to generate negative samples of similar image descriptions based on the original descriptions, followed by synthesizing the corresponding images using text-to-image models. It is well-known that using synthetic data can easily lead to model overfitting. Consequently, various techniques are typically employed to enhance data diversity, such as data augmentation. It would be intuitive to explore whether utilizing multiple models (multiple LLMs and text-to-image models) during the synthesis process might yield better results. I would be interested in seeing an analysis related to this issue, which could make this paper comprehensive.

**Questions:**

See Weaknesses.

**Limitations:**

Yes

---

> ### Author Rebuttal · Authors · 2024-08-06
>
> Thank you for your time and consideration given to our paper. We are delighted that our work is recognized as well-written, easily reproducible, and effective. We appreciate your belief in our approach and its potential to inspire similar strategies in downstream tasks.
>
> In response to your inquiries, please find our clarifications below:
>
> > ## Ablation on Generative Model Choices
>
> While we acknowledge that exploring multiple LLMs and T2I models could be beneficial, our focus in the paper was shaped by three primary considerations:
>
> - The semantic manipulation of text by an LLM is paramount, regardless of the specific LLM used. For better comprehensiveness, **we have performed an additional ablation study**, which is detailed below.
> - T2I models must precisely synthesize images from hard negative captions. Hence, we selected SDXL-Turbo, a state-of-the-art and efficient open-source model, **thus not necessitating further ablation on T2I model choices**.
> - Synthesizing negative captions and images at scale demands significant resources, making holistic evaluations challenging.
>
> For better comprehensiveness, we conducted experiments with different LLMs to generate negative captions, which is a crucial component. The table below presents results from using three LLMs to generate 3M negative captions each for the CC3M dataset. We then trained NegCLIP++ models to assess the effectiveness of these synthetic captions. Notably, Gemma-2 significantly reduces compositionality, while Phi-3 performs best overall. The Phi-3 model was released after the NeurIPS deadline; hence, we use Mistral (second-best choice) in this case.
>
> | Models                      | SugarCrepe | Retrieval (R@5) | ImageNet1k (top-5) |
> |-----------------------------|:------------:|:------------------:|:--------------------:|
> | **Gemma-2b-it**                 | 56.00      | *0.1260*          | **12.09%**             |
> | **Phi-3-mini-4k-instruct**      | *61.22*      | **0.1302**           | *10.94%*             |
> | **Mistral-7b-instruct-v0.2**    | **61.69**      | 0.1072           | 10.52%             |
>
>
> > ## Overfitting Due to Synthetic Data
>
> We agree that synthetic data can lead to overfitting. As noted in our paper and supported by recent work on fully synthetic CLIP models like SynthCLIP [1], three times more synthetic data is required to match the performance of real data due to limited diversity. Our research was motivated by the question: “What kind of synthetic data can enhance performance?” We found that generating hard negative captions and images is the optimal solution. Preliminary analyses on data scaling trends, shown in Figure 3, demonstrate consistent performance improvements with our approach as we scale the synsets.
> While our experiments did not observe overfitting (Figure 3), it remains a potential issue if scaled to billions of hard negative image-text pairs. This topic extends beyond the scope of our current work, and we suggest further exploration by the community in future research.
>
>
> We hope these clarifications address all your questions and enhance the comprehensiveness of our work. Additionally, we provide a summary of responses to other reviews in our global response for your reference.
>
> ---
>
> [1] Hammoud, Hasan Abed Al Kader, Hani Itani, Fabio Pizzati, Philip Torr, Adel Bibi, and Bernard Ghanem. "SynthCLIP: Are We Ready for a Fully Synthetic CLIP Training?." arXiv preprint arXiv:2402.01832 (2024).

---

> > ### Comment · Reviewer_ZTUH · 2024-08-09
> >
> > Thanks for the response. I have no more concerns.

---

> > > ### Author Response · Authors · 2024-08-12
> > > **Reply to Reviewer ZTUH**
> > >
> > > We are pleased to see that all concerns have been effectively addressed.

---

### Author Rebuttal · Authors · 2024-08-07

We sincerely appreciate the constructive feedback provided by the reviewers. It is gratifying to observe the positive evaluations across various dimensions of our work, as highlighted by the reviewers unanimously.

- The reviewers unanimously recognize our paper as **"well-written and easy to follow"** (Reviewers ZTUH, U1Yz, f4Gf, m3ja).
- Reviewers f4Gf and m3ja highlight our work's **originality and significance** in addressing model and data aspects in pre-training CLIP.
- The thoroughly designed experiments, which verify our claims and demonstrate the **effectiveness of TripletCLIP** compared to baselines, are acknowledged by Reviewers f4Gf and m3ja.
- Reviewer ZTUH finds the **paper concise, reproducible, and effective** enough to **encourage wide adoption** of similar strategies across the tasks.
- The **extensive evaluation and consistent improvements** over baselines in compositional and downstream tasks, such as SugarCrepe, zero-shot classification, and image-text retrieval, are noted by Reviewers f4Gf and m3ja.
- Reviewers f4Gf and m3ja **value** the contribution of **releasing large-scale TripletData** to the community.

We have provided detailed responses to each reviewer individually. Below, we summarize responses to two key questions. Additionally, **we have attached the pdf containing detailed benchmark performance** for all the experiments conducted during the rebuttal.

> ## Summarized Responses to Key Questions

- **Choice of LLMs and T2I models:** We provide additional information on the choice of LLMs for generating hard negative captions. We trained NegCLIP++ on 3M generated negative captions for CC3M using three different LLMs and reported the results. We find that Phi-3 performs the best on average, and Gemma-2b surprisingly affects the compositionality significantly. Moreover, unlike LLMs, whose goal is to follow instructions (which is complicated to evaluate), the T2I model's straightforward goal is to synthesize images faithfully. Therefore, we use SDXL Turbo (SOTA fast model) as the default choice without requiring any ablation.


| Models                      |    SugarCrepe    |    Retrieval (R@5)    |    ImageNet1k (top-5)    |
|-----------------------------|:----------------:|:---------------------:|:-----------------------:|
| **Gemma-2b-it**                 |       56.00      |        _0.1260_       |        **12.09%**       |
| **Phi-3-mini-4k-instruct**      |      _61.22_     |       **0.1302**      |         _10.94%_        |
| **Mistral-7b-instruct-v0.2**    |      **61.69**   |        0.1072         |          10.52%         |


- **Comparison with related works:** Reviewer f4Gf kindly pointed out several related works for comparison. We categorize these related works into two categories: 1) **Orthogonal works:** Our paper makes orthogonal contributions to methods like CyCLIP, meaning our work can be jointly utilized with these works, advocating independent contributions that do not necessitate comparisons. 2) **Relevant works focusing on hard negatives:** Most of the existing works are orthogonal and focus on various ways to improve hard negative captions with minor perturbations in loss functions. While two relevant works focus on generating negative images.
    - In response to Reviewer f4Gf, we have performed additional finetuning-related experiments and reported the detailed comparisons. We find that additional baselines struggle even to match the compositionality performance of NegCLIP. While **TripletCLIP retains the competitive compositionality performance.**

Again, we thank the reviewers and AC for their time reviewing our paper and providing detailed feedback. We hope that we have addressed all remaining concerns and questions. We look forward to the rebuttal discussion period.

---

### Comment · Area_Chair_m6zB · 2024-08-12
**Please read the author rebuttal, other reviews and respond to the authors NOW!**

Dear Reviewers,

Thanks to those of you who already responded to the authors acknowledging the rebuttal and asking follow-up questions if any.

Those who have not responded yet, please do the following ASAP: thoroughly read the rebuttal, the other reviews and respond to the authors about whether all your questions / concerns have been addressed or not. If not, please elaborate on which questions / concerns are still not addressed so that the authors have fair chance of addressing them before the author-reviewer discussion period ends in ~41 hours from now (August 13th, 11:59pm AoE).

Your AC

---

### Decision · Program_Chairs · 2024-09-25

**Decision:**

Accept (poster)

**Comment:**

The reviewers find the proposed method of generating image and text hard negatives to be effective and the resulting generated dataset to be a useful resource for the community. The reviewer also appreciate the extensive and meticulous experimentation and find the experimental results to be convincing. The reviewers had raised some concerns, but the rebuttal successfully addressed most of them and all reviewers recommend acceptance. The authors are recommended to improve the final paper version by following the reviewer recommendations.